# IGGT: Instance-Grounded Geometry Transformer for Semantic 3D Reconstruction

**Hao Li**[1,2,3,*], **Zhengyu Zou**[1,*], **Fangfu Liu**[4], **Xuanyang Zhang**[3*], **Fangzhou Hong**[2],
**Yukang Cao**[2], **Yushi Lan**[2], **Manyuan Zhang**[5], **Gang Yu**[3], **Dingwen Zhang**[1✉], **Ziwei Liu**[2]
[1]NWPU    [2]S-Lab, NTU    [3]StepFun, Inc.    [4]THU    [5]MMLab, CUHK

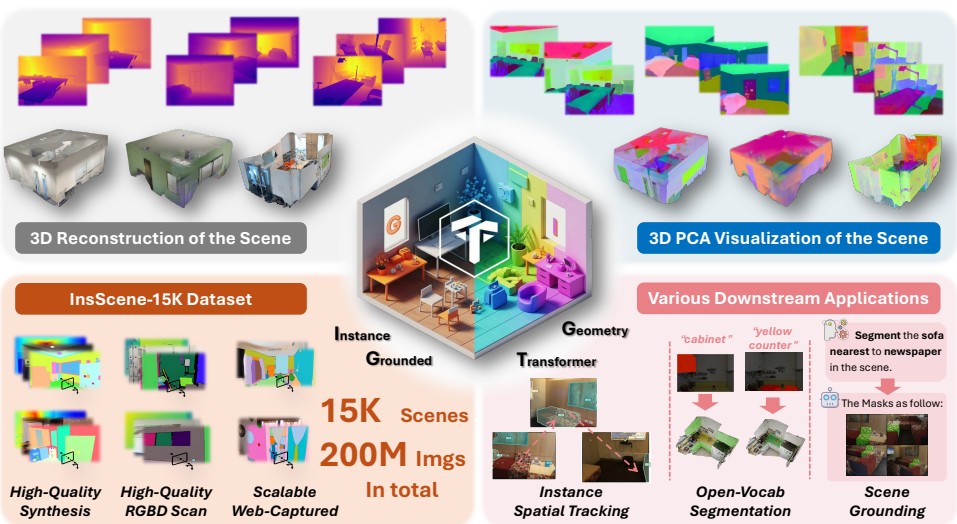

Figure 1: **IGGT**: building upon our curated large-scale dataset InsScene-15K, we propose a novel end-to-end framework that enables geometric reconstruction and contextual understanding in a unified representation. This paradigm facilitates a wide range of applications, including multi-view instance matching, 2D / 3D open-vocabulary segmentation, and scene grounding.

## Abstract

Humans naturally perceive the geometric structure and semantic content of a 3D world as intertwined dimensions, enabling coherent and accurate understanding of complex scenes. However, most prior approaches prioritize training large geometry models for low-level 3D reconstruction and treat high-level spatial understanding in isolation, overlooking the crucial interplay between these two fundamental aspects of 3D-scene analysis, thereby limiting generalization and leading to poor performance in downstream 3D understanding tasks. Recent attempts have mitigated this issue by simply aligning 3D models with specific language models, thus restricting perception to the aligned model's capacity and limiting adaptability to downstream tasks. In this paper, we propose **Instance-Grounded Geometry Transformer (IGGT)**, an end-to-end large unified transformer to unify the knowledge for both spatial reconstruction and instance-level contextual understanding. Specifically, we design a *3D-Consistent Contrastive Learning* strategy that guides IGGT encode a unified representation with geometric structures and instance-grounded clustering through only 2D visual inputs. This representation supports consistent lifting of 2D visual inputs into a coherent 3D scene with explicitly distinct object instances. To facilitate this task, we further construct **InsScene-15K**, a large-scale dataset with high-quality RGB images, poses, depth maps, and 3D-consistent instance-level mask annotations with a novel data curation pipeline. Unlike previous methods that bound with a specific language model, we introduce an *Instance-Grounded Scene Understanding* paradigm, where instance masks serve as the bridge connecting our

---

*Equal Contribution, *Project Leader. ✉Corresponding Authors.

unified representation with diverse Visual Language Models (VLMs) in a plug-and-play manner, substantially expanding downstream understanding capabilities. Extensive experiments on multi-view instance matching, open-vocabulary segmentation, and QA scene grounding demonstrate that IGGT outperforms state-of-the-art methods in both quality and consistency for semantic 3D reconstruction. `https://github.com/lifuguan/IGGT_official`.

# 1 INTRODUCTION

A foundational goal in the pursuit of spatial intelligence (Yang et al., 2025) is to build representations that mirror human understanding—capturing both the precise geometric structure and rich semantic content of a scene from visual sensory inputs such as RGB images. Such representations are vital for enabling downstream tasks like robotic manipulation (Qu et al., 2025), AR / VR (Jiang et al., 2025), and planning (Zhang et al., 2024).

Previous methods (Zust et al., 2025; Fan et al., 2024; Sun et al., 2025) tackle this challenge through a fragmented paradigm, decoupling 3D geometric reconstruction and high-level semantic understanding into isolated tasks. Typically, they first leverage geometry-focused techniques (*e.g.,* Multi-View Stereo (MVS) methods (Schönberger et al., 2016; Schönberger & Frahm, 2016) or off-the-shelf large Image-to-3D models (Wang et al., 2024; 2025)) to predict low-level 3D structures, followed by vision-language models (VLMs) (Bai et al., 2023; 2025) or 2D segmentation models (Cheng et al., 2022) to perform high-level semantic segmentation tasks. However, these disjointed approaches are inherently flawed, as they propagate errors between stages and fail to leverage the mutual context between shape and identity, preventing them from enhancing each other's capabilities and hindering their ability to support model reconstruction.

Recently emerged methods (Fan et al., 2024; Sun et al., 2025) attempt to bridge this gap by aligning spatial models with specific VLM (Li et al., 2022). However, these approaches suffer from three critical limitations. First, since 3D geometry contains low-level, fine-grained structural signals, forcing a strict alignment with high-level textual concepts can over-smooth the representation, degrading high-frequency geometric details and undermining multi-view consistency. Second, this tight coupling to a specific VLM architecture inherently restricts the performance to the base model (*e.g.*, LSeg (Li et al., 2022)) and prevents the integration of newer, more powerful foundation models (*e.g.*, CLIP (Radford et al., 2021), SigLIP (Tschannen et al., 2025)). Third, since these VLMs (Li et al., 2022; Ghiasi et al., 2022) are mainly trained on 2D image–text pairs, their aligned features often fail to distinguish objects within the same semantic category, which significantly limits more downstream applications (*e.g.,* , 3D instance-consistent tracking under large viewpoint changes and spatial QA when interfaced with VLMs).

To address this, we propose **I**nstance-**G**rounded **G**eometry **T**ransformer (**IGGT**), a novel end-to-end framework that unifies the representation for spatial reconstruction and contextual understanding. Instead of simply aligning geometry with language features, our key idea is to *couple both factors by joint training and encourage the model to autonomously learn the relationship between 3D instance-level semantics and their geometric structures, yielding **mutual improvements** in contextual understanding and geometry reconstruction*. Specifically, **1)** we employ a large Unified Transformer to encode multi-view images into unified token representations of the 3D scene, which are decoded by a Geometry Head and an Instance Head into geometric point maps and an instance clustering fields, respectively. **2)** we employ a cross-modal fusion block with a window-shifted attention mechanism, enabling the Instance Head to leverage fine-grained geometric features at pixel level to enhance its spatial awareness. **3)** To further improve multi-view consistency of the instance fields, we design a 3D-consistent contrastive learning strategy that guides IGGT to learn both geometric structures and instance-grounded clustering features. As instance-level geometry-semantics aligned annotations remain scarce in the community, we facilitate this task by presenting a large-scale dataset coined **InsScene-15K**, a meticulously constructed dataset comprising high-quality RGB images, poses, depth maps, and 3D-consistent instance masks.

**One more thing**, after training the full model (*i.e.,* IGGT), we design an *Instance-Grounded Scene Understanding* strategy, where instance masks serve as the bridge connecting IGGT with diverse VLMs. Such a paradigm not only enables the seamless, plug-and-play integration of various vision-language models (VLMs) such as CLIP and SigLIP to lift downstream task performance, but also

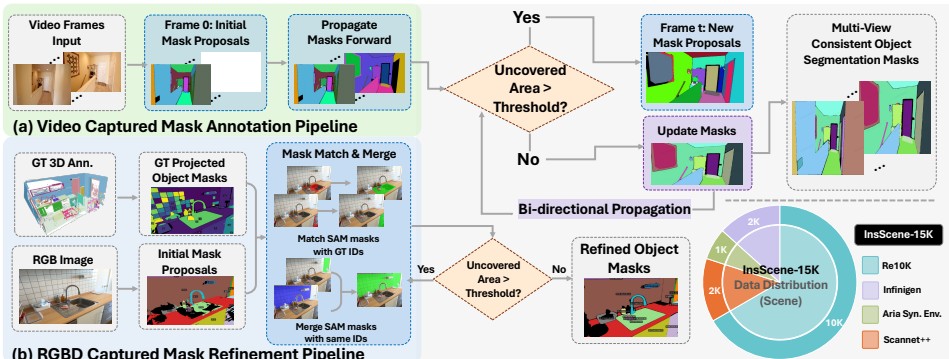

Figure 2: **Data Curation Pipeline.** Our data is collected from various sources and then annotated by a novel data engine driven by SAM2 (Ravi et al., 2024). (a) For video captured scenes (*i.e.*, RE10k (Zhou et al., 2018)), we annotate them through a customized SAM2 video dense prediction pipeline. (b) For RGBD-scan scenes (e.g., ScanNet++ (Yeshwanth et al., 2023)), we regenerate dense mask annotations for each image and align them with the projected coarse GT masks.

extends to Large Multimodal Models (LMMs) (Bai et al., 2023; 2025), unlocking more sophisticated scene understanding and a broader spectrum of applications like scene grounding.

We validate our framework through extensive experiments on diverse downstream tasks (*e.g.,* multi-view instance matching, open-vocabulary segmentation, and scene grounding), demonstrating its superiority over state-of-the-art methods in both task performance and 3D scene coherence.

## 2  INSSCENE-15K DATASET

We construct the InsScene-15K dataset (in Sec. 2), where each scene includes corresponding RGB images, depth maps, poses, and 3D-consistent instance segmentation masks. To maintain consistency, we ensure that each instance retains a unique ID across all views.

Our data curation pipeline systematically integrates three distinct categories of data to ensure comprehensiveness and diversity, as illustrated in Fig. 2: 1) synthesis (Aria (Pan et al., 2023), Infinigen (Raistrick et al., 2024)); 2) Video captured (RE10K (Zhou et al., 2018)); 3) RGBD captured (ScanNet++ (Yeshwanth et al., 2023)). For synthetic datasets (*e.g.*, Aria and Infinigen), we simultaneously generate the RGB image, depth map, camera pose, and object-level segmentation masks for each rendered view. Since the simulation environment provides perfectly accurate 2D ground-truth masks (in Fig. 3 (a)), we use them directly without any post-processing. Moreover, regarding real-world scenarios, we propose a novel data curation pipeline that includes multi-view mask an-

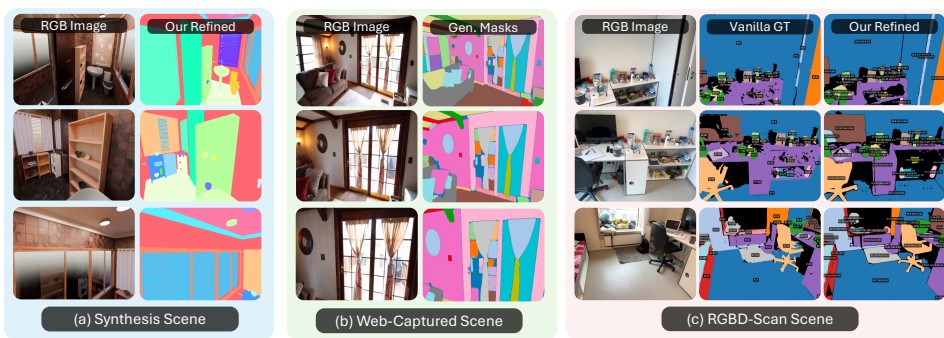

Figure 3: Visualization of mask annotations from three different sources. For the RGBD-scan scene, we additionally compare the vanilla ground-truth masks from ScanNet++ (Yeshwanth et al., 2023) with our refined annotations, along with their corresponding matched IDs and mIoU scores.

notation and refinement stages, driven by SAM2 (Ravi et al., 2024). Specifically, for real-world video-captured scenes such as RE10K (Zhou et al., 2018) (Fig. 2a), our method first employs SAM to generate dense mask proposals on the initial frame. These proposals are then used as prompts for the SAM2 video object segmenter to propagate masks temporally throughout the sequence. To handle new objects and mitigate drift, we adopt an iterative strategy that designates a new keyframe whenever the unsegmented area increases, where SAM is reapplied to discover objects in the uncovered regions. After processing the entire video, a final bi-directional propagation pass ensures high temporal consistency across object tracks. This curation strategy provides scalable and diverse annotations that enhance the generalization ability of our model.

For challenging datasets with large-scale camera motion but coarse 3D annotations such as Scan-Net++ (Yeshwanth et al., 2023), we first project the 3D annotations into 2D to obtain initial image-level object masks. While this guarantees multi-view consistency of object IDs, the masks are often coarse and imprecise. To improve their quality, we use SAM2 to generate fine-grained initial mask proposals that are accurate in shape but lack identity information. These proposals are then aligned with the projected ground-truth masks to assign consistent object IDs (Fig. 3 (c)), and proposals belonging to the same ID are merged into complete masks. The process is iteratively refined until all image regions are covered. This pipeline (Fig. 2 (b)) achieves both multi-view ID consistency and shape-accurate annotations, substantially improving 2D mask quality for real-world scenarios.

## 3 METHODOLOGY

### 3.1 OVERVIEW

Our method consists of two main phases. Firstly, we propose IGGT (in Sec. 3.2), a unified foundation model that simultaneously predicts instance-discriminative features at the spatial level and performs 3D reconstruction through 3D-consistent contrastive learning on large-scale datasets. Secondly, we propose an instance-grounded scene understanding strategy (Sec. 3.3). This strategy employs unsupervised clustering to partition the scene into instances by grouping the predicted features into masks with consistent instance IDs. These masks are then used to guide state-of-the-art vision-language models (VLMs, e.g., CLIP, OpenSeg) and large multimodal models (LMMs, e.g., GPT-4o, Qwen2.5-VL) to perform open-vocabulary scene querying and grounding tasks.

### 3.2 ARCHITECTURE OF IGGT

As illustrated in Fig. 4, given $N$ input images $\{I^i \in \mathbb{R}^{H \times W \times 3}\}_{i=1}^N$, we aim to forge a unified representation, enabling comprehensive 3D reconstruction and understanding in a mutually reinforcing manner. Specifically, we propose IGGT $\mathcal{F}$, which predicts camera parameters $t_i$, depth map $D_i$, point map $P_i$, and 3D-consistent, instance-level feature maps $S_i$ in a feed-forward manner:

$$\mathcal{F} : \{I_i\}_{i=1}^N \mapsto (t_i, D_i, P_i, S_i)_{i=1}^N. \tag{1}$$

Our IGGT consists of three parts: 1) a Large Unified Transformer to capture Unified Token Representation from multiple images; 2) two Downstream Heads with a Cross-Modal Fusion Block to simultaneously predict geometric structures and corresponding instance features; 3) a 3D consistent supervision to empower the model to construct 3D-consistent feature fields of the scenes.

**Large Unified Transformer.** We follow VGGT to construct a 1B parameter large unified Transformer, designed to encode the multi-view images $\{I_i\}_{i=1}^N$ into a set of powerful unified token representations $\{\mathbf{T}_i \in \mathbb{R}^{M \times D}\}_{i=1}^N$, where $M$ denotes the numbers of the tokens for each image and $D$ is the dimension of the token. Our large Unified Transformer first adopts pretrained DINOv2 (Oquab et al., 2023) to extract patch-level image tokens. To support arbitrary multi-view inputs while maintaining permutation equivariance, a learnable camera token is concatenated to each view's token sequences. Subsequently, 24 blocks of intra-view self-attention and global-view cross-attention are applied to transform the image tokens into unified tokens $\{\mathbf{T}_i\}_{i=1}^N$, capturing both local and global context, which enables a holistic and globally consistent understanding of the 3D scene.

**Downstream Heads and Cross-Modal Fusion Block.** We employ two downstream branches—Geometry Head and Instance Head—to decode the unified tokens $\mathbf{T}_i\}$ into geometric and instance features, respectively. The Geometry Head, inheriting its design from VGGT, is composed of three distinct modules: a camera predictor, a depth predictor, and a point predictor. The

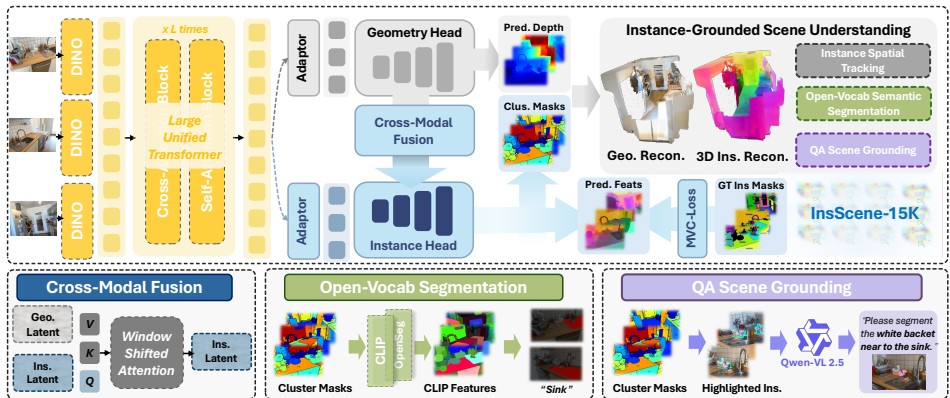

Figure 4: Overview of **IGGT**. Given input images, our method encodes them into a series of Unified Token Representations, which are then processed by the Geometry Head and the Instance Head to produce high-quality geometric reconstructions and instance-grounded clusterings simultaneously. In the end, we introduce Instance-Grounded Scene Understanding to perform multiple applications.

camera predictor is tasked with regressing camera parameters, including extrinsics and intrinsics, from camera-specific tokens. For dense prediction, the depth and point predictors employ a DPT-like architecture (Ranftl et al., 2021). This architecture reconstructs a hierarchical geometric feature $\boldsymbol{F}_i^{pt} = \{F_{i,(l)}^{pt}\}_{l=1}^4$ from the unified tokens through progressive upsampling and multi-scale fusion network $\Phi_{pt}(\cdot)$. Similar to this dense prediction paradigm, our Instance Head $\Phi_{ins}(\cdot)$ also adopts a DPT-like architecture to perform dense instance features $\boldsymbol{F}_i^{ins} = \{F_{i,(l)}^{ins}\}_{l=1}^4$:

$$\{F_i^{pt}\} = \Phi_{pt}(\{\mathbf{T}_i\}), \quad \{F_i^{ins}\} = \Phi_{ins}(\{\mathbf{T}_i\}). \tag{2}$$

Moreover, to enhance the fine-grained spatial awareness of the instance head, we propose a cross-modal fusion block $\mathcal{F}_{\text{win}}(\cdot)$, which utilizes a sliding window cross attention to embed spatial structure into the instance representation, making them more sensitive to object boundaries and spatial layouts while avoiding the quadratic complexity of global attention:

$$\hat{F}_{i,(l)}^{ins} = F_{i,(l)}^{ins} + \mathcal{F}_{\text{win}}(Q = F_{i,(l)}^{ins}, K = F_{i,(l)}^{pt}, V = F_{i,(l)}^{pt}). \tag{3}$$

After that, we concatenate all refined instance features $\{\hat{F}_{i,(l)}^{ins}\}$ and map them through a conventional $3 \times 3$ convolutional layer to 8 dimensional instance features $O_{ins} \in \mathbb{R}^{N \times 8 \times H \times W}$.

**3D-Consistent Contrastive Supervision.** We enforce 3D consistency on the instance features $O_{\text{ins}} \in \mathbb{R}^{N \times 8 \times H \times W}$ by applying a multi-view contrastive loss $\mathcal{L}_{mvc}$, which is designed to pull features from the same 3D instance together across views while pushing features from different instances apart. Given a set of sampled pixels $\mathcal{P}$, the loss is formulated as:

$$\mathcal{L}_{mvc} = \lambda_{pull} \cdot \sum_{\substack{p_i, p_j \in \mathcal{P} \\ m(p_i) = m(p_j)}} d(f_{p_i}, f_{p_j}) + \lambda_{push} \cdot \sum_{\substack{p_i, p_j \in \mathcal{P} \\ m(p_i) \neq m(p_j)}} \max(0, M - d(f_{p_i}, f_{p_j})) \tag{4}$$

Here, $d(\cdot, \cdot)$ is the L2 distance between normalized features, $m(p_i)$ is the instance ID of pixel $p_i$. The coefficients $\lambda_{\text{pull}}$ and $\lambda_{\text{push}}$ balance the pulling and pushing terms, while $M$ is a margin hyper-parameter that controls the discriminative between different instances. This objective structures the instance representations according to the 3D scene geometry, improving generalization. Overall, we train the whole model in a multi-task loss:

$$\mathcal{L}_{overall} = \mathcal{L}_{pose} + \mathcal{L}_{depth} + \mathcal{L}_{pmap} + \mathcal{L}_{mvc}, \tag{5}$$

where geometry supervision terms pose $\mathcal{L}_{pose}$, depth $\mathcal{L}_{depth}$, and point map $\mathcal{L}_{pmap}$ are followed by the training paradigm of VGGT, which is used to supervise the outputs of the geometry head.

## 3.3 INSTANCE-GROUNDED SCENE UNDERSTANDING

Unlike prior approaches that are tightly coupled with a specific language model (e.g., for Open-Vocabulary Segmentation) and thus limited to a single type of task, we decouple our framework from

Table 1: Quantitative Results on ScanNet (Dai et al., 2017). Here we showcase the capability overview and report the performance of multi-view instance matching (MV Ins. Mat.), reconstruction, and 2D / 3D open-vocabulary semantic segmentation. The **bold** denotes the best results.

| Model | Capability | | | MV Ins. Mat. | | Recon. Metric | | Open-Vocab. Semantic Segment | | |
|---|---|---|---|---|---|---|---|---|---|---|
| | Recon. | Understand | Mat. | T-mIoU↑ | T-SR↑ | Abs. Rel↓ | $\tau$↑ | 2D mIoU↑ | 2D mAcc↑ | 3D mIoU↑ |
| LSeg | ✗ | ✓ | ✗ | - | - | - | - | 58.11 | 65.76 | - |
| OpenSeg | ✗ | ✓ | ✗ | - | - | - | - | 42.33 | 68.06 | - |
| NeRF-DFF | ✓ | ✓ | ✗ | - | - | 7.99 | 36.53 | 45.40 | 65.29 | 12.29 |
| Feature-3DGS | ✓ | ✓ | ✗ | - | - | 6.48 | 41.63 | 57.69 | 63.26 | 23.42 |
| LSM (2 Views) | ✓ | ✓ | ✗ | - | - | 4.22 | 58.65 | 53.07 | 53.86 | - |
| LSM (Multi-Views) | ✓ | ✓ | ✗ | - | - | 3.17 | 64.81 | 53.40 | 59.50 | 35.37 |
| SpaTracker+SAM | ✗ | ✗ | ✓ | 26.43 | 38.57 | - | - | - | - | - |
| SAM2* | ✗ | ✓ | ✓ | 53.74 | 71.25 | - | - | - | - | - |
| VGGT | ✓ | ✗ | ✗ | - | - | **1.84** | 83.60 | - | - | - |
| Ours | ✓ | ✓ | ✓ | **69.41** | **98.66** | 1.90 | **83.71** | **60.46** | **81.84** | **39.68** |

Table 2: Quantitative Results on ScanNet++ (Yeshwanth et al., 2023). Here we report the multi-view instance matching quality, reconstruction accuracy, and 2D / 3D open-vocabulary semantic segmentation accuracy.

| Model | MV Ins. Mat. | | Recon. Metric | | Open-Vocab. Semantic Segment | | |
|---|---|---|---|---|---|---|---|
| | T-mIoU↑ | T-SR↑ | Abs. Rel↓ | $\tau$↑ | 2D mIoU↑ | 2D mAcc↑ | 3D mIoU↑ |
| LSeg | - | - | - | - | 22.61 | 34.42 | - |
| OpenSeg | - | - | - | - | 13.92 | 48.13 | - |
| Feature-3DGS | - | - | 5.92 | 41.64 | 22.47 | 33.14 | 10.59 |
| LSM (2 Views) | - | - | 4.22 | 74.02 | 17.76 | 26.95 | - |
| LSM (Multi-Views) | - | - | 2.96 | 83.28 | 17.88 | 27.84 | 15.17 |
| SpaTracker+SAM | 16.15 | 23.68 | - | - | - | - | - |
| SAM2* | 44.16 | 57.89 | - | - | - | - | - |
| VGGT | - | - | 2.75 | 85.41 | - | - | - |
| Ours | **73.02** | **98.90** | 2.61 | **85.66** | **31.31** | **70.78** | **20.14** |

specific language models and propose a novel Instance-Grounded Scene Understanding strategy to support a broad range of downstream tasks. As shown in Tab. 1, our method is the only one that simultaneously enables multi-view instance matching, image-to-3D reconstruction, and scene understanding, while achieving state-of-the-art performance across all tasks.

**Multi-View Instance Matching.** Specifically, inspired by SAMPart3D (Liu et al., 2025), we apply the density-based clustering algorithm HDBSCAN (McInnes et al., 2017) that gathers multi-view 2D instance features $\{O_i^{ins}\}$ into $K$ distinct clusters, where each cluster represents a unique object instance present in the scene. Then we re-project the assigned cluster labels to their corresponding pixel locations produces a set of 3D-consistent 2D instance masks $\{M_{i,k}^{ins}\}_{k=1}^K$. Such a paradigm enables dense tracking and segmentation of specific instances across multi-view images by leveraging explicit 3D priors, in stark contrast to existing methods that are either limited to discriminating category-level features or lose targets during significant camera motion.

**Open-Vocabulary Semantic Segmentation.** These 3D-consistent instance masks serve as effective prompts for any off-the-shelf VLMs (Radford et al., 2021; Ghiasi et al., 2022), enabling them to perform robust open-vocabulary semantic segmentation by assigning a semantic category to each mask-defined region. Here we take OpenSeg (Ghiasi et al., 2022) as an example. It first produces image-wise features $\{F_i^{lang} \in \mathbb{R}^{D \times H \times W}\}_{i=1}^N$, which considers contextual information to enable accurate visual-language alignment of the features. We then aggregate the features within each 2D instance mask $\{\mathbf{f}_k^{lang} \in \mathbb{R}^D\}_{i=1}^K$ via average mask pooling, yielding a compact representation for each instance. This step not only integrates the mask priors into the visual-language space, but also sharpens object boundaries and captures fine-grained local category cues, making the subsequent semantic assignment more accurate and robust.

**QA Scene Grounding.** Unlike prior methods that directly align 3D features with language embeddings, our approach offers greater flexibility by decoupling instance clusterings, which can then interact with LMMs (Bai et al., 2025; Team et al., 2023) to support object-centric QA in 3D scenes. Concretely, as shown in Fig. 4, given $N$ views, we highlight the image regions corresponding to the same instance $k$ with masks $\{M_{i,k}^{ins}\}_{i=1}^N$ (rendered in red), and query the LMM with yes/no questions to verify object consistency across views. Finally, we aggregate all positive ("yes") responses and concatenate the corresponding masks to form the final segmentation output.

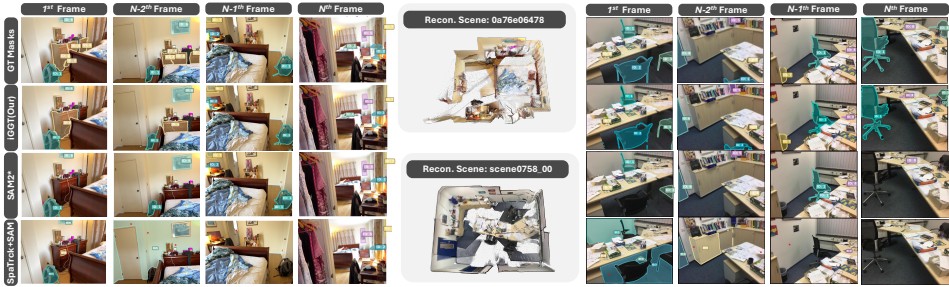

Figure 5: Qualitative results on multi-view instance matching. We present two example scenes from ScanNet (Dai et al., 2017) and ScanNet++ (Yeshwanth et al., 2023), and compare our method with SAM2* and SpaTracker+SAM. All instances are visualized with distinct IDs and colors for clarity.

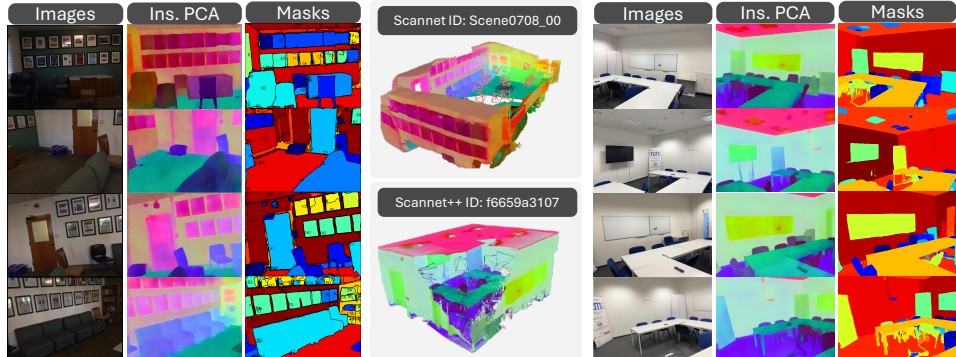

Figure 6: We visualize our 3D-consistent PCA results with corresponding clustered masks derived from instance-grounded features. Similar colors in PCA indicate higher feature similarity between instances. For clustered masks, the same object instance shares the same color across multi-views.

## 4 EXPERIMENTS

**Evaluation Details.** We conduct comprehensive experiments on the ScanNet (Dai et al., 2017) and ScanNet++ (Yeshwanth et al., 2023) datasets. From each dataset, we randomly select 10 scenes and sample 8-10 images per scene, with the selection strategy designed to maximize spatial coverage of the scene while preserving sufficient overlap to ensure cross-view consistency. (a) For *Multi-View Instance Matching* evaluation, we evaluate tracking performance using Temporal mIoU (T-mIoU) and Temporal Success Rate (T-SR). T-mIoU measures the segmentation accuracy of the same object across different views, while T-SR assesses whether the object is successfully tracked in every view. (b) For *Open-Vocabulary Segmentation* evaluation, we follow LangSplat (Qin et al., 2024) and LangSurf (Li et al., 2024b), which adopt mIoU and mAcc to measure 2D segmentation accuracy. In addition, we evaluate the 3D mIoU metric by aligning the reconstructed scene with the ground-truth point cloud. (c) For *Reconstruction* evaluation, we follow LSM (Fan et al., 2024) and VGGT (Wang et al., 2025) that utilize Absolute Relative Error (Abs. Rel) and Inlier Ratio ($\tau$) with a threshold of 1.03 to assess each scene. The details of these metrics are shown in the appendix.

**Evaluation of Multi-View Instance Matching.** To comprehensively evaluate the tracking quality of our proposed method and competing approaches, particularly under large viewpoint changes with multiple objects, we manually annotate a subset of objects across several scenes with precise ground-truth labels (more visualization in the Appendix). For baseline methods, we modify SAM2 (Ravi et al., 2024) to support dense segmentation and tracking under multi-view inputs, denoted as SAM2*. In addition, we integrate SAM into SpaTrackerV2 (Xiao et al., 2025), where tracking points are used as prompts to perform dense segmentation. Tab. 1 and Tab. 2 present the quantitative results, demonstrating the significant superiority of our method. By leveraging implicit 3D reasoning, our approach successfully distinguishes object identities to achieve nearly 100% T-SR accuracy. In contrast, baseline methods fail at this crucial task, yielding a T-mIoU below 30%, whereas our approach surpasses 60%. This performance gap is visually demonstrated in Fig. 5, where our method successfully tracks and segments the chair under large camera motions, while competing methods lose the track.

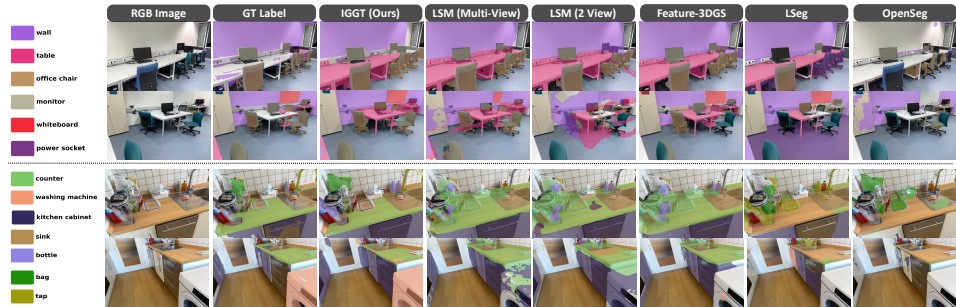

Figure 7: Qualitative Results of 2D Open-Vocabulary Segmentation on ScanNet and ScanNet++.

Furthermore, we provide additional visualizations of our 3D-consistent instance features using Principal Component Analysis (PCA), along with their corresponding clustered masks, as shown in Fig. 6. As illustrated, IGGT produces 3D-consistent instance-grounded features that remain discriminative across multiple views: multiple instances with the same category exhibit similar yet distinguishable colors in the PCA space. This property serves as a crucial foundation for the Multi-View Instance Matching task, as it enables consistent tracking and segmentation of individual objects even under large motions and in the presence of many similar instances.

**Evaluation of Open-Vocabulary Segmentation.** We compare our method with other Image-to-3D feedforward method (Fan et al., 2024), per-scene optimized methods (Zhou et al., 2024; Kobayashi et al., 2022), and 2D methods (Ghiasi et al., 2022; Li et al., 2022) on both ScanNet and Scanent++ datasets. The results are reported in Tab. 1 and Tab. 2. On ScanNet++, our method achieves leading performance, surpassing other approaches by 8.34% in mIoU for segmentation and 7.88% in mAcc for object localization. This performance improvement is attributed to our method's superior multi-view consistency, which helps correct object recognition errors caused by incomplete views, as illustrated in Fig. 7, where the sink is difficult to identify due to limited viewpoint coverage. On the other hand, we also evaluate the accuracy of depth estimation on multi-view

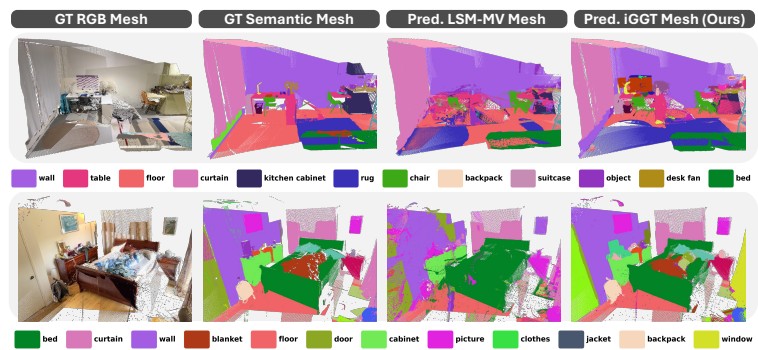

Figure 8: Visualization of 3D Open-Vocabulary Segmentation.

inputs. The results show that our method is on par with VGGT on ScanNet, and outperforms VGGT on ScanNet++ by 0.14 in Abs. Rel and 0.25 in $\tau$, benefiting from the mutual enhancement of semantics and geometry achieved through joint training. This performance improvement is further demonstrated in 3D segmentation (see Tab. 1 and Tab. 2), where our method outperforms previous approaches by 4.31% and 4.97% in terms of 3D mIoU. As shown in Fig. 8, our method produces more consistent segmentations within the same regions and significantly fewer floaters in 3D, which indicates that our approach yields superior 3D semantic representations.

**Class-Agnostic Mask Segmentation.** Following recent 3D instance segmentation works (Yin et al., 2024; Yang et al., 2024; Huang et al., 2024c), we provide a class-agnostic instance segmentation evaluation on the ScanNet dataset, comparing our method against VGGT + Graph Cut and VGGT + SAI3D (Yin et al., 2024). We report AP25, AP50, and AP in Tab. 3.

Our method outperforms VGGT+Graph Cut by 8.83 AP while avoiding its expensive mesh generation, reducing runtime by about 8 minutes. Moreover, it approaches the performance of the per-scene optimization method VGGT+SAI3D with nearly a 5× speed-up (2.5 min *v.s.* 12.2 min), demonstrating the efficiency of our uni-

Table 3: Comparison of class-agnostic segmentation with different methods on AP metrics.

| Method | AP | $AP_{50}$ | $AP_{25}$ | Time |
|---|---|---|---|---|
| VGGT + Graph Cut | 3.42 | 9.30 | 30.86 | 10.65min |
| VGGT + SAI3D | 14.94 | 31.06 | 50.07 | 12.22min |
| Ours | 12.25 | 24.93 | 47.55 | 2.52min |

Table 4: Comparison of runtime for different methods.

| Method | SfM Time | Per-Scene Time | Infer. Time | Post Proc. Time | Final Time |
|---|---|---|---|---|---|
| NeRF-DFF | 50.33s | 3min | - | - | 3.84min |
| Feature-3DGS | 50.33s | 47min | - | - | 47.84min |
| LSM (Multi-Views) | - | - | 15.98s | 13.72s | 29.70s |
| VGGT+Graph Cut | - | - | 0.426s | 10.64min | 10.65min |
| VGGT+SAI3D | - | - | 0.426s | 12.21min | 12.22min |
| Ours | - | - | 0.545s | 2.51min | 2.52min |

fied reconstruction and instance understanding framework. These improvements are also supported by the qualitative results in Fig. 9. More analysis and settings are shown in the appendix.

**Runtime Analysis.** Here, we use 10 images from a single scene to evaluate the detailed runtime, as reported in Tab. 4. Compared with NeRF/3D-GS or other per-scene optimization methods, our approach substantially reduces reconstruction time. In contrast, feed-forward methods such as LSM (Multi-View) achieve faster processing but suffer from limited reconstruction accuracy and are unable to support tasks like instance matching or scene QA. More in the appendix.

**Out-of-Distribution (OOD) Results.** We evaluate our model on various OOD scenarios: outdoor scenes (ETH3D (Schops et al., 2017)), autonomous driving scenes (Waymo Open Dataset (Sun et al., 2020)), and egocentric-view data (robotics data and a self-collected dataset). Qualitative results in Fig. 10 show that our method can reconstruct scenes and produce 3D-consistent instance clustering on these unseen domains, demonstrating strong generalization ability. More in the appendix.

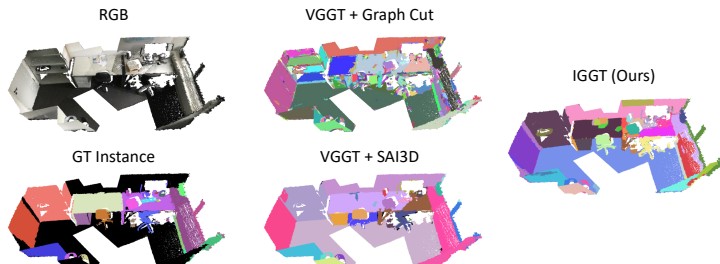

Figure 9: Visualization of the Class-Agnostic 3D Mask Segmentation Results.

**Applications of QA Scene Grounding.** We present the QA application results in Fig. 11 on the Teatime scene from the LERF-OVS (Kerr et al., 2023) dataset, and compare our approach against the state-of-the-art Gemini 2.5 Pro (Comanici et al., 2025). As shown, our instance-grounded querying fully leverages the reasoning capacity of LMMs, achieving accurate segmentation for complex prompts and superior multi-view consistency compared to existing unified generation–understanding models, thereby enabling more complex QA tasks in 3D scenes.

**Ablation Study.** We show the training curve of IGGT in Fig. 13. Without the cross-modal fusion model, the instance head struggles to capture high-resolution geometric information, leading to more difficult convergence, as reflected in the sharpness of the chair edges in the PCA visualization. We also conduct ablations on integrating different VLMs into our method (*e.g.*, LSeg (Li et al., 2022), CLIP (Radford et al., 2021), OpenSeg (Ghiasi et al., 2022)). As shown in the table, LSeg and OpenSeg, with better global context representation, achieve higher accuracy on background classes (e.g., cabinet), while CLIP, with superior text alignment, performs better on complex categories such as 'DALL-E' and 'Ottolegnghi' in Fig. 12. This further demonstrates the flexibility of our method in using different VLMs to achieve improved text query performance.

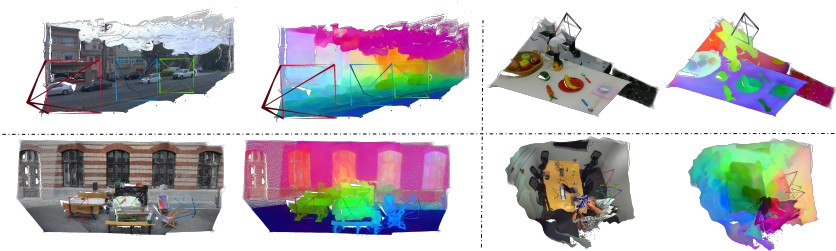

Figure 10: Visualization of OOD scenes (*e.g.*, outdoor, autonomous driving, egocentric, robotics).

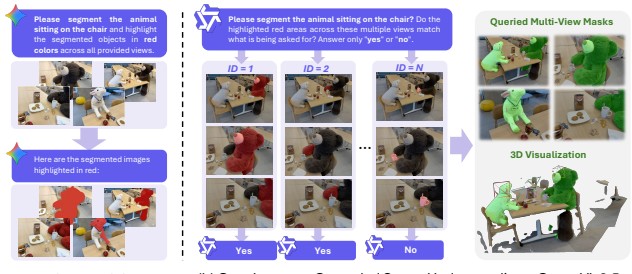

(a) Vanilla Gemini 2.5 Pro    (b) Ours Instance-Grounded Scene Understanding + Qwen-VL 2.5

Figure 11: Applications of QA Scene Understanding compared with vanilla Gemini 2.5 Pro Comanici et al. (2025).

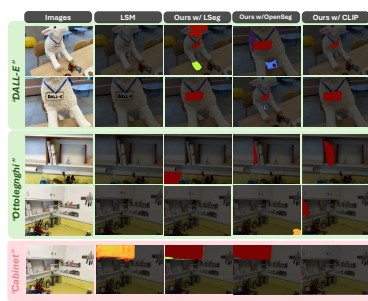

Figure 12: Visualization of our method using different VLMs.

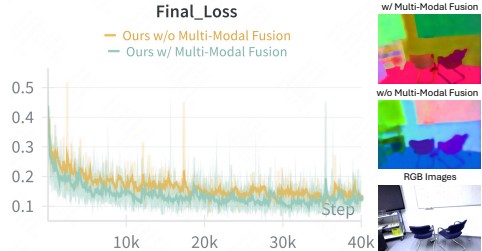

Figure 13: Ablation on Cross-Modal Fusion.

Table 5: Integration with Different VLMs.

| Method | ScanNet | | ScanNet++ | |
|---|---|---|---|---|
| | mIoU↑ | mAcc↑ | mIoU↑ | mAcc↑ |
| Ours *w/* Lseg | **60.46** | **81.84** | 22.72 | 63.56 |
| Ours *w/* CLIP | 49.36 | 62.68 | 21.52 | 61.36 |
| Ours *w/* OpenSeg | 58.12 | 78.75 | **31.31** | **70.78** |

## 5 RELATED WORK

**Spatial Foundation Models.** Image-to-3D reconstruction has evolved from early SfM pipelines like COLMAP (Schonberger & Frahm, 2016), which estimate camera poses and sparse point clouds, to more advanced methods like 3D Gaussian Splatting (3DGS) (Kerbl et al., 2023) for efficient novel view synthesis. Scene Representation Transformers (Sajjadi et al., 2022) represent images as latent tokens, enabling view synthesis without accurate poses, but still struggle with explicit geometry and generalization. DUSt3R (Wang et al., 2024) improves upon this by directly regressing dense point maps from unposed image pairs, while VGGT (Wang et al., 2025) scales this approach to multiple images with competitive accuracy. However, these methods remain focused on geometric reconstruction, often neglecting higher-level scene understanding.

**3D Scene Understanding.** Integrating semantics into 3D reconstruction is vital for scene understanding. LLM-based methods (Huang et al., 2024b; Yu et al., 2025) inject 3D priors into LMMs, whereas 3D-based methods (Peng et al., 2023; Li et al., 2025) align VLM features in 3D space for understanding, which relies on pre-reconstructed 3D scenes. Meanwhile, methods such as LangSplat (Qin et al., 2024) inject vision-language features into 3D Gaussian Splatting to enable semantic reasoning, but they typically require dense multi-view inputs and per-scene optimization. Moreover, feed-forward approaches such as Panst3R (Zust et al., 2025) and DUSt3R (Wang et al., 2024) attempt feed-forward scene understanding, but decouple geometry and semantics, limiting mutual benefits. Methods like LSM (Fan et al., 2024) and Uni3R (Sun et al., 2025) align spatial models with vision-language models (e.g., LSeg (Li et al., 2022)), but face limitations in integrating stronger VLMs and struggle with fine-grained, instance-level queries in complex scenes.

## 6 CONCLUSION

In this paper, we introduce IGGT, a novel end-to-end framework that unifies the representation for both spatial reconstruction and contextual understanding in a 3D scene. The key to our success is that we couple geometric and instance-level semantic features by joint training and unleash the potential of a unified large transformer to achieve mutual improvements in contextual understanding and geometry reconstruction. To facilitate this task, we further present a large scale dataset called InsScene-15K, including high-quality RGB images, poses, depth maps, and 3D-consistent instance masks. Moreover, our proposed instance-grounded scene understanding strategy enables IGGT with plug-and-play integration of various VLMs and LMMs, unlocking a broader range of applications.

# 7 ETHICS STATEMENT

This work focuses on improving spatial reconstruction and understanding. While our model is trained on self-annotated datasets based on standard open-source images and tested in controlled settings, we acknowledge that any AI system may potentially exhibit biases or produce unexpected behaviors. Our research is intended for academic exploration only, and we emphasize that any such outcomes do not reflect the views of the authors. We support the development of AI technologies that are ethical, safe, and aligned with societal values.

# 8 REPRODUCIBILITY STATEMENT

All code and model checkpoints will be publicly released to ensure reproducibility.

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

# A APPENDIX

## A.1 ACKNOWLEDGMENT

This research is supported by cash and in-kind funding from NTU S-Lab and industry partner(s). This study is also supported by the Ministry of Education, Singapore, under its MOE AcRF Tier 2 (MOE-T2EP20221-0012, MOE-T2EP20223-0002). This work was supported in part by the National Natural Science Foundation of China under Grant 62293543, Grant 62322605, and Grant 625B2148.

## A.2 USE OF LARGE LANGUAGE MODELS

Large Language Models (LLMs) are used exclusively for minor grammar corrections and stylistic polishing of the manuscript. They are not involved in the design of the methodology, execution of experiments, analysis of results, or any other aspect of the scientific contribution.

## A.3 RELATED WORK

**Spatial Foundation Model** Image-to-3D reconstruction is a long-standing problem in computer vision. Early pipelines such as COLMAP (Schonberger & Frahm, 2016) and related SfM methods estimate camera poses and sparse point clouds, often followed by multi-view stereo (MVS) to obtain dense geometry (Yao et al., 2018). Building on such SfM-based initialization, 3D Gaussian Splatting (3DGS) (Kerbl et al., 2023) introduced a highly efficient representation for photorealistic novel view synthesis, inspiring reconstruction-oriented extensions (Chen et al., 2024; Huang et al., 2024a). To reduce the reliance on accurate calibration, Scene Representation Transformers (Sajjadi et al., 2022; Li et al., 2024a) represent multiple images as latent scene tokens, enabling novel view synthesis under uncertain or missing poses, though they still struggle to produce explicit geometry and generalize reliably. DUSt3R (Wang et al., 2024) takes a further step by directly regressing dense point maps from unposed image pairs, achieving pixel-aligned geometry without SfM initialization. In contrast, VGGT (Wang et al., 2025) scales this paradigm to dozens to hundreds of images in a single feed-forward pass, jointly predicting cameras, depth, point maps, and tracks with competitive accuracy to optimization-based pipelines. Despite these advances, these methods remain focused on low-level geometric reconstruction while overlooking higher-level scene understanding.

**3D Scene Understanding** Integrating semantics into 3D reconstructions is crucial for higher-level scene understanding tasks. Existing LLM-based methods (Huang et al., 2024b; Yu et al., 2025) inject 3D priors into LMMs and finetune them to perform 3D grounding. However, these approaches rely on pre-reconstructed 3D scenes. Recent efforts (Zhang et al., 2025; Li et al., 2024c; Zhou et al., 2024; Li et al., 2024b; Qin et al., 2024) like LangSplat (Qin et al., 2024) inject vision-language features (e.g., CLIP (Radford et al., 2021)) into 3D Gaussian Splatting, enabling semantic reasoning over reconstructed scenes. However, these methods typically require dense multi-view inputs and per-scene optimization, which hinders scalability. More generalizable approaches like Panst3R (Zust et al., 2025) build on DUSt3R (Wang et al., 2024) to achieve feed-forward 3D scene understanding directly from posed or unposed images. Yet, they often decouple reconstruction from understanding and freeze the geometry module, which restricts mutual benefits between the two and leads to suboptimal semantic grounding. Parallel attempts such as LSM (Fan et al., 2024) and Uni3R (Sun et al., 2025) seek to bridge geometry and semantics by aligning spatial models with specific vision-language models (e.g., LSeg (Li et al., 2022)), but this tight coupling has two key drawbacks: (1) it prevents seamless integration of stronger VLMs (Tschannen et al., 2025; Siméoni et al., 2025) as they emerge, thereby constraining text query performance; (2) the alignment is typically at the category level rather than instance-level, so these methods struggle with fine-grained, object-centric QA in scenes that contain multiple similar instances.

To address this problem, our proposed framework, IGGT, addresses these limitations by learning a unified representation for both reconstruction and understanding. Instead of tightly coupling with a single VLM, we introduce an instance-grounded paradigm where instance masks serve as a bridge to connect with diverse VLMs and Large Multimodal Models (LMMs) in a plug-and-play manner, substantially expanding downstream capabilities.

## A.4 TRAINING DETAILS

Our model is initialized with weights from VGGT (Wang et al., 2025) and fine-tuned on the InsScene-15K dataset, which contains 15,000 scenes. Training is performed on 8 NVIDIA A800 GPUs for 2 days using the AdamW optimizer. The learning rate is set to $1 \times 10^{-6}$ for the large unified Transformer backbone and $1 \times 10^{-5}$ for both the geometry and instance heads. For each training batch, we randomly sample 1–12 frames from a randomly selected scene, yielding a total of 24 images per batch. For hyper-parameter settings, we set $\lambda_{pull} = 2.0$, $\lambda_{pull} = 1.0$ and $M = 1.0$.

## A.5 METRICS FOR DIFFERENT TASKS

**Multi-View Instance Matching**. For the Multi-View Instance Matching task, we evaluate tracking performance using Temporal mIoU (T-mIoU) and Temporal Success Rate (T-SR). Given an object $o$ and its predicted masks $\{\hat{M}_t^o\}_{t=1}^T$ across $T$ views with corresponding ground-truth masks $\{M_t^o\}_{t=1}^T$, T-mIoU is defined as

$$\text{T-mIoU}(o) = \frac{1}{T} \sum_{t=1}^{T} \frac{|\hat{M}_t^o \cap M_t^o|}{|\hat{M}_t^o \cup M_t^o|}.$$

T-SR evaluates whether the object is successfully tracked across all views, and is defined as

$$\text{T-SR}(o) = \mathbb{1}\left[\forall t \in \{1, \ldots, T\}, \ |\hat{M}_t^o| > 0\right],$$

where $\mathbb{1}[\cdot]$ denotes the indicator function. The final scores are averaged over all objects in the dataset.

**3D Semantic Segmentation mIoU**. To evaluate 3D semantic segmentation, we first obtain the RGB 3D points from per-image point maps and align them with the ground truth. Next, we assign semantic labels to the corresponding 3D points based on the results of 2D open-vocabulary segmentation. These labeled 3D points are subsequently voxelized, and the 3D mIoU is computed based on the voxel representation. Fig. 15 illustrates the overall pipeline. Additionally, Fig. 16 presents qualitative results of 3D open-vocabulary segmentation. For LSM, based on its two-view input, we apply the global alignment strategy of Dust3R to optimize the point maps across all views. For Feature-3DGS, the ground-truth point maps are used as the initial input. However, due to the sparsity of input views, its reconstruction quality remains limited.

## A.6 ADDITION INFORMATION OF OUR INSSCENE-15K DATASET

Fig. 17 presents the vanilla masks and the refined counterparts, together with the IDs that establish the correspondence between them. The refined masks contain fewer unannotated regions and align more closely with the actual objects. Training with these high-quality instance-level masks facilitates more accurate instance-level segmentation and tracking. Meanwhile, we provide a dataset card to highlight the uniqueness of our dataset, as shown in Table 6. While web-captured datasets such as RE10K (Zhou et al., 2018) offer a large number of scenes (i.e., high diversity), they lack depth and 3D-consistent instance masks. For RGB-D datasets like ScanNet (Dai et al., 2017) and ScanNet++ (Yeshwanth et al., 2023), although they provide accurate depth measurements, they only annotate coarse 2D masks (as shown in Fig. 17) and contain only around 1,000 scenes. For synthetic datasets like Infinigen (Raistrick et al., 2024), although they provide precise instance masks, most object assets are reused with relatively simple structures, which also leads to limited diversity. As for our InsScene-15K dataset, we build upon our own annotation pipeline to re-annotate ScanNet++ and RE10K with instance masks, ensuring that our dataset provides high-quality poses, depth, and instance masks at scale (around 15,000 scenes).

## A.7 ABLATION OF 3D-CONSISTENT CONTRASTIVE SUPERVISION

As shown in Tab. 7, we perform an ablation study on ScanNet++ dataset regarding the weighting coefficient $\lambda$ of the contrastive loss $\mathcal{L}_{\text{mvc}}$, with the weight of the geometric losses fixed to 1. Specifically, we scale $\lambda$ by factors of $\times 2$ and $\times 10$ to analyze its impact on performance. Here, our method maintains comparable performance under the $\times 2$ setting (i.e., $\lambda = 0.5, 2$ ), demonstrating the robustness of the overall training procedure. In contrast, under the $\lambda = 0.1$ setting, the contrastive

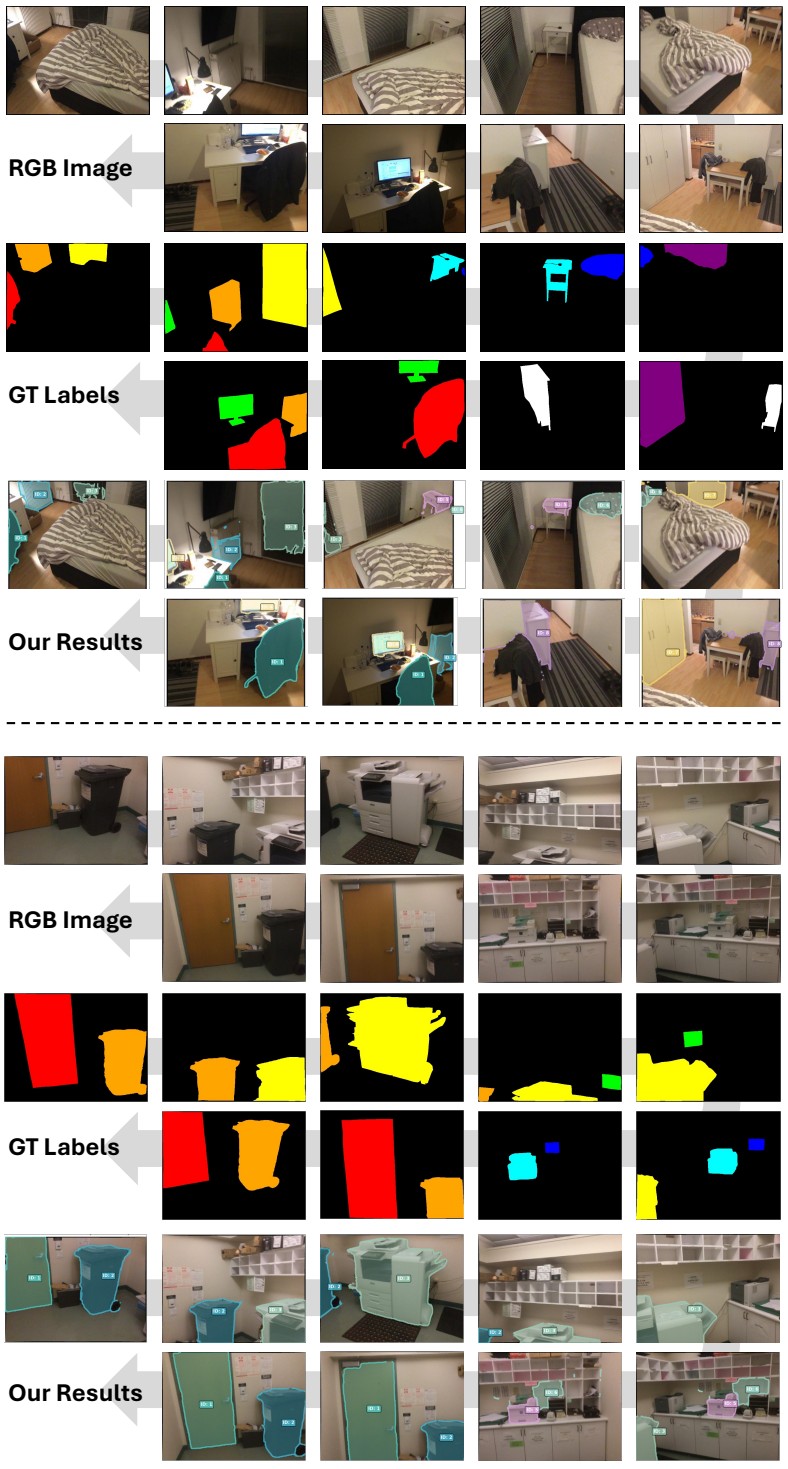

Figure 14: Visualization of our manually annotated tracking GT and our tracking results.

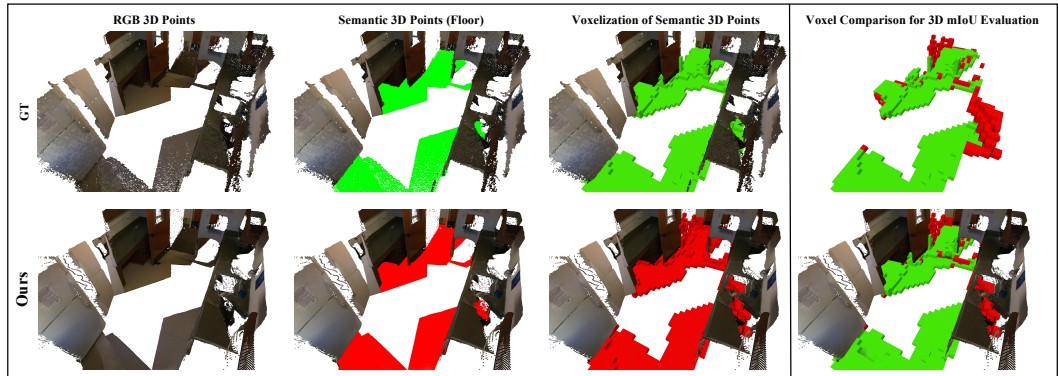

Figure 15: Visualization of the pipeline from RGB 3D points to semantic labeling, voxelization, and voxel comparison for 3D mIoU.

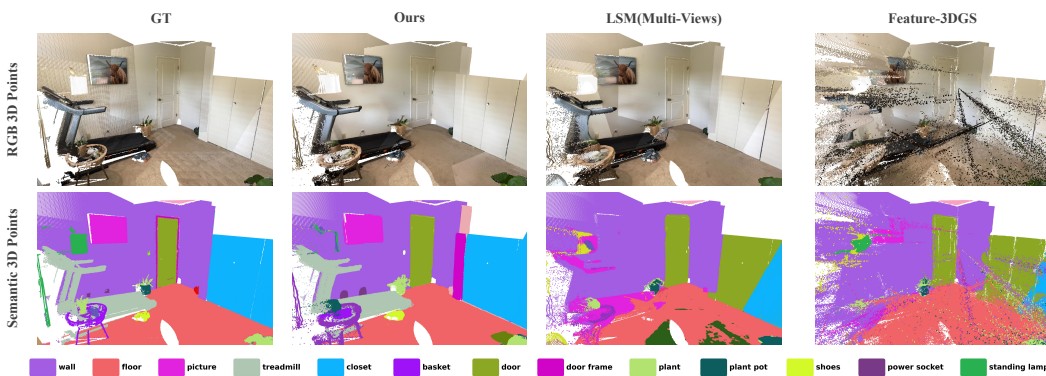

Figure 16: We visualize the RGB and semantic 3D points of the ground truth, IGGT(Ours), LSM(Multi-Views), and Feature-3DGS.

supervision fails to provide sufficiently discriminative instance features, leading to degraded mask quality in the post-processing clustering stage, resulting a big drop in understanding performance (10.44% in 2D mIoU). Meanwhile, under the $\lambda = 10$ setting, the contrastive supervision becomes dominant and weakens the geometric supervision, leading to degraded geometric reconstruction performance (a 6.28% drop in $\tau$) as well as a decline in 3D segmentation accuracy (a 4.96% drop in 3D mIoU).

Table 6: **Comparison of Different Datasets.** Here, we evaluate these datasets along five dimensions: RGB images, camera poses, depth, instance masks, and diversity. Datasets with good-quality annotations in a given dimension are marked with '✓', while those without such annotations, or with low-quality ones, are marked with '✗'.

| Dataset | RGB | Pose | Depth | Instance Masks | Diversity |
|---|---|---|---|---|---|
| RE10K | ✓ | ✓ | ✗ | ✗ | ✓ |
| ScanNet | ✓ | ✓ | ✓ | ✗ | ✗ |
| ScanNet++ | ✓ | ✓ | ✓ | ✗ | ✗ |
| Infinigen | ✓ | ✓ | ✓ | ✓ | ✗ |
| InsScene-15K (Ours) | ✓ | ✓ | ✓ | ✓ | ✓ |

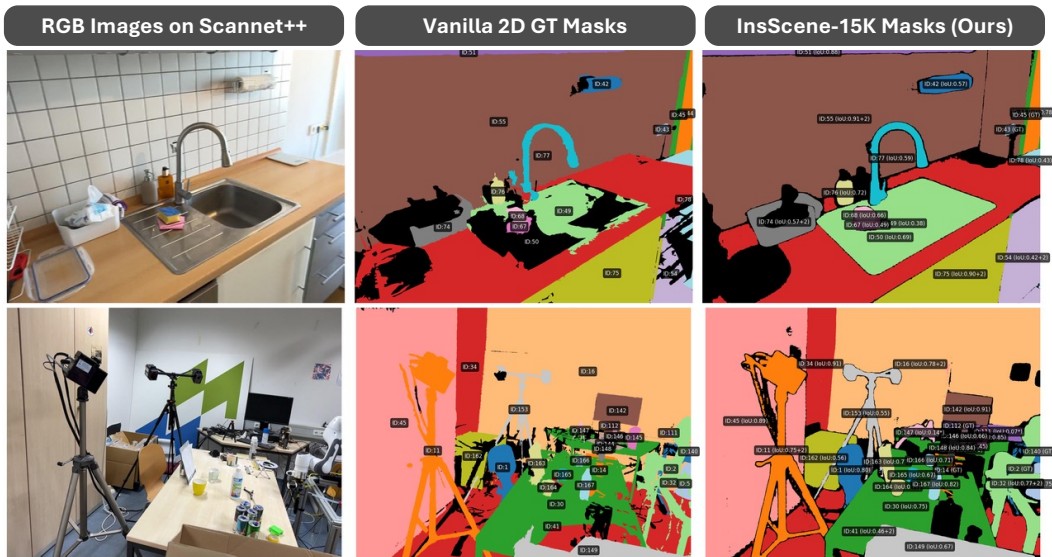

Figure 17: Comparison between vanilla ScanNet++ GT masks and our refined results.

Table 7: Ablation study with different values of $\lambda$ of contrastive supervision.

| Metrics | $\lambda$ | | | | |
|---|---|---|---|---|---|
| | 0.1 | 0.5 | 1 | 2 | 10 |
| 2D mIoU | 20.63 | 31.07 | 31.31 | 31.86 | **32.60** |
| 3D mIoU | 13.89 | 19.79 | **20.14** | 20.01 | 15.18 |
| $\tau$ | 85.35 | 85.43 | **85.66** | 85.19 | 79.38 |

### A.8 CLASS-AGNOSTIC SEGMENTATION EXPERIMENTS

Following recent 3D instance segmentation works (Yin et al., 2024; Yang et al., 2024; Huang et al., 2024c), we have also provided a class-agnostic instance segmentation evaluation on the ScanNet dataset, comparing our method against VGGT + Graph Cut and VGGT + SAI3D (Yin et al., 2024). For VGGT + Graph Cut, we first use VGGT to reconstruct the entire scene as a point cloud from the input images, then compute normal vectors for the points and generate a mesh using Poisson reconstruction (Kazhdan et al., 2006). Finally, following SAI3D, we apply a normal-based graph cut algorithm (Felzenszwalb & Huttenlocher, 2004) to over-segment the mesh into segmentation results (superpoints). For VGGT + SAI3D, the pipeline is directly built on VGGT + Graph Cut, and the predicted depth, poses, and superpoints are then fed into SAI3D to obtain the final class-agnostic instance segmentation results.

We follow the ScanNet instance segmentation benchmark to evaluate class-agnostic instance segmentation. First, we use the ground-truth depth and camera poses to project the ground-truth 2D instance masks into point clouds, obtaining the ground-truth class-agnostic 3D instance masks. Subsequently, we align the point clouds predicted by IGGT and VGGT with the ground-truth point clouds, and then apply nearest-neighbor matching to establish a one-to-one correspondence between the predicted and ground-truth class-agnostic 3D instance masks. Additionally, we followed the practice of SAI3D by ignoring instances representing 'wall' and 'floor'.

We report the average precision scores at IoU thresholds of 0.25 (AP25) and 0.50 (AP50), as well as averaged over IoU thresholds from 0.50 to 0.95 in increments of 0.05 (AP) in Tab. 3. Our method significantly outperforms graph-based grouping approaches such as VGGT+Graph Cut across all metrics, achieving an 8.83 improvement in AP. In Fig. 18, we present the RGB point clouds, the ground-truth class-agnostic 3D instance segmentation, as well as the class-agnostic 3D instance segmentation results predicted by VGGT+Graph Cut, VGGT+SAI3D, and IGGT(Ours), allowing

for a direct visual comparison. Meanwhile, since this graph cut algorithm requires a mesh as input, a substantial amount of time is spent on mesh generation, whereas our approach can directly output multi-view consistent instance features for clustering, reducing the overall runtime by approximately 8 minutes. Moreover, our method even approaches the performance of the per-scene optimization method VGGT+SAI3D, while achieving nearly a 5× reduction in runtime (2.5 min *v.s.* 12.2 min). This further showcases the effectiveness of our unified framework, which jointly learns geometric reconstruction and instance understanding.

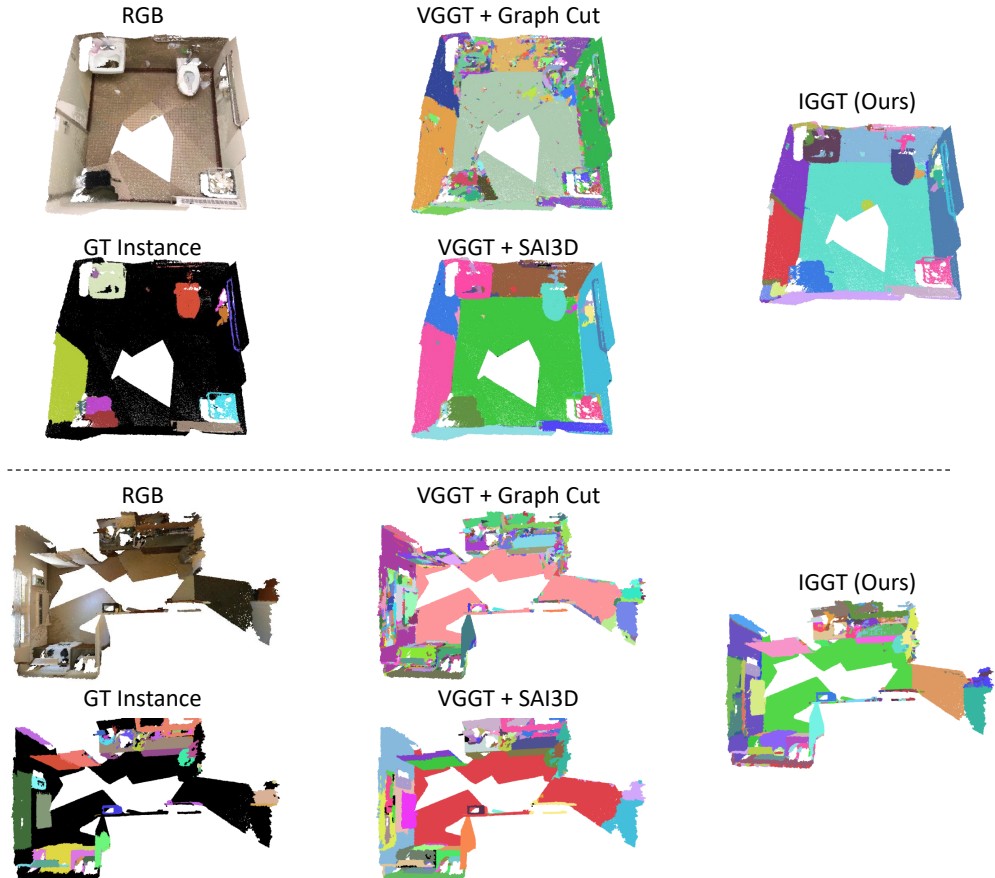

Figure 18: Additional Visualization of the RGB point clouds, the ground-truth 3D instance segmentation (where black indicates ignored instances), and the predicted 3D instance segmentation from VGGT+Graph Cut, VGGT+SAI3D, and IGGT (Ours).

### A.9 RUNTIME ANALYSIS.

Here, we use 10 images from a single scene to evaluate the detailed runtime, as reported in Tab. 4.

Open-Vocabulary Semantic Segmentation. For NeRF and 3DGS methods, they require massive time consumption for per-scene optimization. Moreover, they also require camera poses as input, which are typically estimated using COLMAP. For the LSM (Multi-Views) method, building on its two-view inference, we further feed all images into Dust3R to infer the camera parameters and point maps for each view, and then apply the global alignment strategy of Dust3R to optimize the point maps across all views. Although it takes only about 30 seconds, its open-vocabulary semantic segmentation performance is inferior to ours, and it is unable to produce class-agnostic instance segmentation results.

Class-Agnostic Instance Segmentation. For the VGGT+Graph Cut method, we adopt the normal-based graph cut algorithm provided by the ScanNet official code. We compute normal vectors for the point clouds predicted by VGGT and generate a mesh using Poisson reconstruction. These additional steps are time-consuming and introduce significant overhead. For the VGGT+SAI3D method, which is built upon the VGGT+Graph Cut pipeline, a substantial portion of the runtime is similarly spent on mesh processing. For our method, most of the runtime is spent on the HDBSCAN clustering step, which takes about two minutes. Nevertheless, our approach remains the second fastest among all compared methods.

### A.10 ADDITIONAL VISUALIZATION ON GRANULARITIES

Since our method does not support prompt-based inputs like SAM or Semantic-SAM, it is unable to directly predict instance features at different granularities. However, we can still obtain instance masks at different scales by adjusting the parameters of HDBSCAN. As illustrated in Fig. 19, we showcase three different clustering granularities, where the red circle areas are the discriminative region.

| RGB | PCA | Small | Medium | Large |

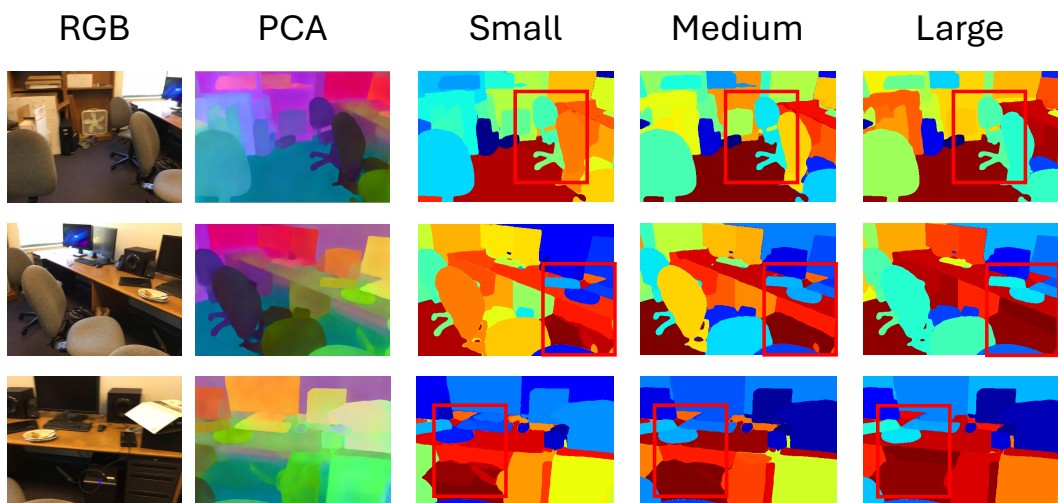

Figure 19: Visualization on clustered masks with different granularities.

### A.11 ADDITIONAL VISUALIZATION ON 3D VQA

As shown in Fig. 20, we showcase two tasks, object counting and spatial relation reasoning, derived from ScanRefer (Chen et al., 2020) and ScanQA (Azuma et al., 2022). When provided with the masks generated by our method, Gemini 2.5 Pro successfully completes both tasks with a high success rate. In contrast, the success rate drops significantly without our masks, highlighting the importance and effectiveness of explicit spatial parsing for general 3D scene understanding.

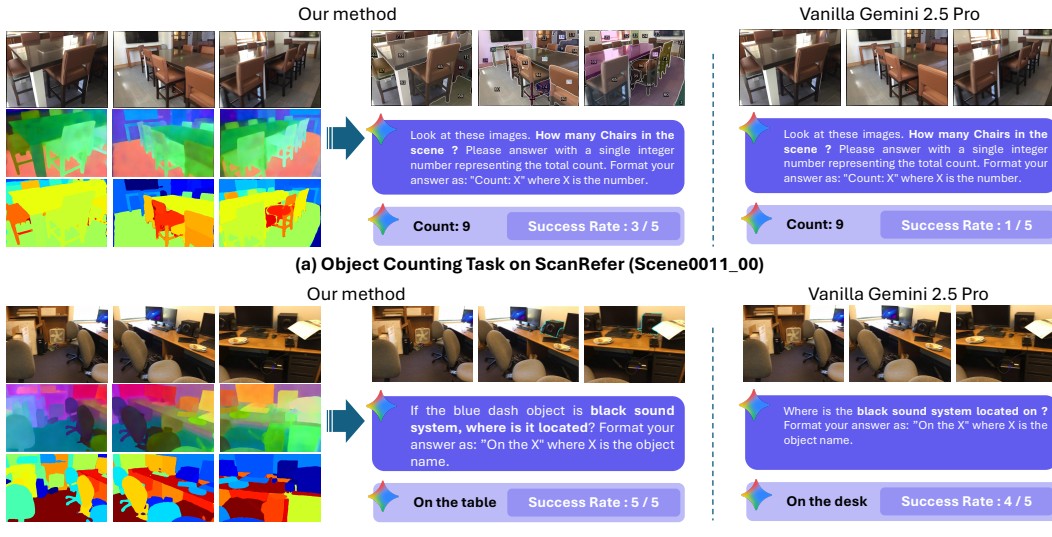

Figure 20: Demonstration of 3D VQA tasks sampled from ScanRefer (Chen et al., 2020) and ScanQA (Azuma et al., 2022).

## A.12 OUT-OF-DISTRIBUTION (OOD) RESULTS

To demonstrate the generalization capability of our model, we evaluated our model on: Outdoor scenes (ETH3D (Schops et al., 2017)), Autonomous driving scenes (Waymo Open Dataset (Sun et al., 2020)), Egocentric-view data (robotics data and a self-collected data), and low-light scenario. These visualization results are shown in the appendix. As shown, our method is able to correctly handle outdoor scenes (Fig. 21, Fig. 22), low-light scene (Fig. 23), and even capture short-term dynamic motion in both robotic manipulation and egocentric scenarios (Fig. 24(a)). In the robotic scenario, the motion of the robotic arm during the grasping sequence is correctly tracked (blue mask), while in the egocentric scenario, the cardboard box is also consistently tracked (blue mask). These diverse test cases demonstrate the robustness of our method, especially considering that it is trained only on static scenes. However, as shown in Fig. 24(b), for the fisheye scenario in egocentric data, our method, similar to VGGT, encounters challenges in producing geometrically accurate reconstructions due to the more complex intrinsic parameters of fisheye cameras.

## A.13 LIMITATION

Our method adopts an unsupervised clustering strategy on the proposed Instance-Grounded Clustering for post-processing. As a result, the accuracy of object boundaries in the clustered masks cannot yet rival that of state-of-the-art segmentation models (e.g., SAM2 (Ravi et al., 2024)). Future work may integrate stronger DETR-based (Cheng et al., 2022) instance heads and larger annotated datasets to improve segmentation accuracy.

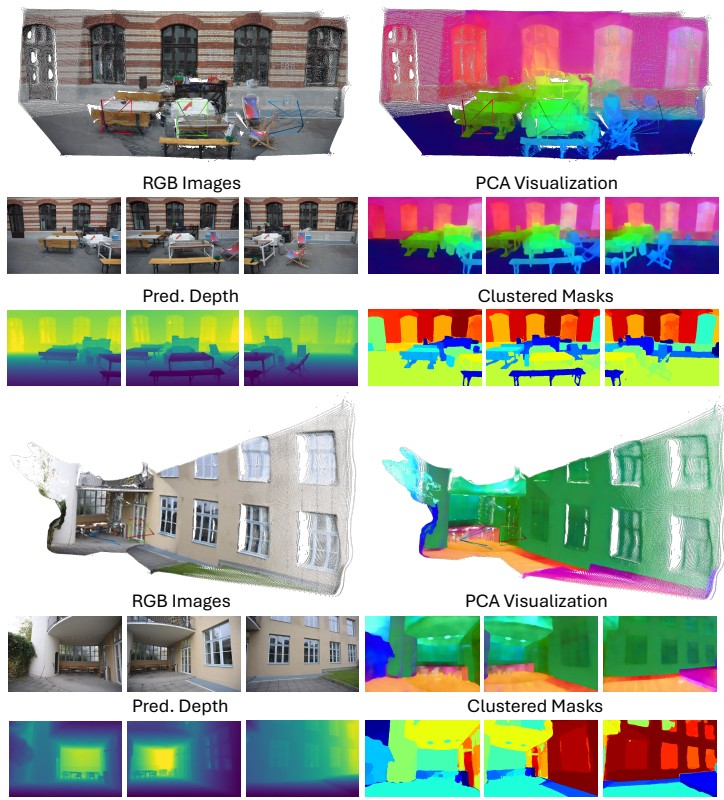

Figure 21: Visualization of outdoor scenes from the ETH3D dataset (Schops et al., 2017).

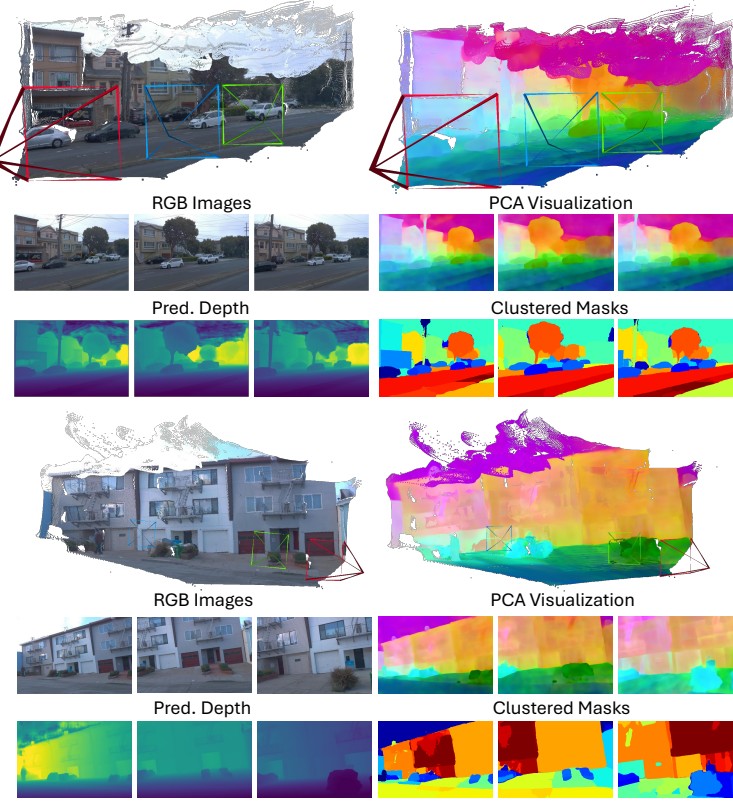

Figure 22: Visualization of autonomous driving scenes from the Waymo Open Dataset (Sun et al., 2020).

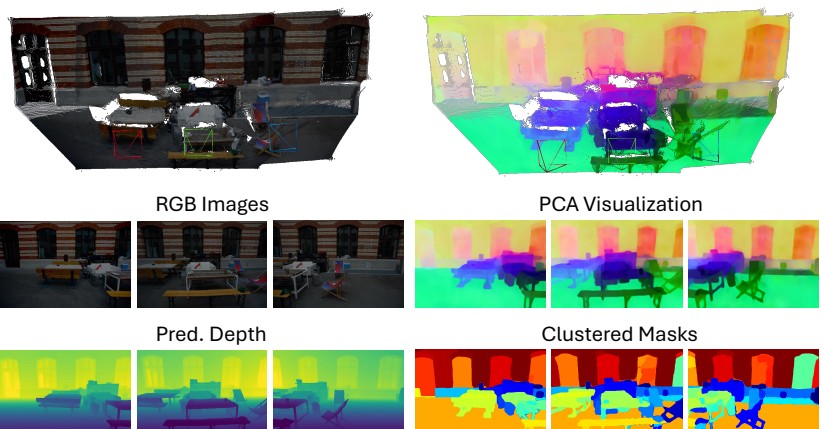

Figure 23: Visualization of a low-light scenario. For comparison, the corresponding original scene under normal lighting conditions is provided in Fig. 21.

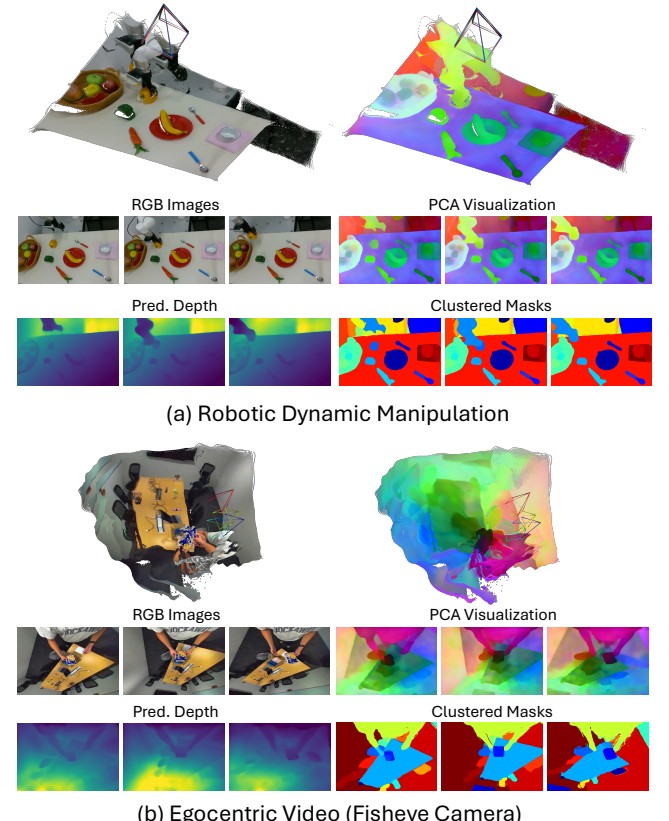

Figure 24: Visualization of dynamic data, including a robotic scenario and an egocentric scenario (with a fisheye camera).

