# OpenReview forum: "IGGT: Instance-Grounded Geometry Transformer for Semantic 3D Reconstruction"
_ICLR.cc/2026/Conference — ICLR 2026 Poster_

### Official Review · Reviewer_Jjwe · 2025-10-25

**Soundness:** 4
**Presentation:** 4
**Contribution:** 4
**Rating:** 8
**Confidence:** 5

**Summary:**

The paper presents IGGT, a unified transformer that consumes multi-view RGB and jointly predicts geometry (cameras, depth, point maps) and instance-grounded features suitable for open-vocabulary 2D/3D semantics and downstream multi-modal reasoning. A shared backbone drives two heads (Geometry / Instance). A window-shifted cross-modal fusion injects geometric context into the instance stream, and a 3D-consistent contrastive objective aligns instance identity across views. The authors also curate InsScene-15K, a collection with multi-view images, poses, depths, and multi-view-consistent instance masks assembled via a practical SAM/SAM2-based workflow.

The contribution can be summarized:
1. A unified transformer with shared tokens and two heads (Geometry / Instance).
2. A window-shifted cross-modal fusion to inject geometric cues into the instance stream.
3. A 3D-consistent contrastive loss to keep instance identities aligned across views.
4. InsScene-15K: practical multi-view data with pose/depth and multi-view-consistent instance masks (SAM/SAM2-based curation).
5. Competitive results on ScanNet and ScanNet++, plus an easy interface to VLM/LMM tools via instance masks.

**Strengths:**

1. The dataset contribution in this 3D understanding community is pretty essential. InsScene-15K is a practical, geometry-aware, multi-view dataset with view-consistent instance IDs that scales via SAM-based curation, supports tracking/OVS/reconstruction in one place, and cleanly interfaces with VLM/LMM grounding, filling a real gap for unified 3D perception.

2. Following VGGT in the geometry representation, IGGT uses multi-view images encoder to unifythe visual token represeentations. Additionally, it added an instance head to output the dense instance features. A Cross-Modal Fusion Block is used to fulfill this downstream head.

3. The results are genuinely promising: on ScanNet/ScanNet++ it meets or exceeds strong baselines in instance spatial tracking (T-mIoU/T-SR), open-vocabulary 2D/3D segmentation, and geometric reconstruction, with clean qualitative masks and stable depth. The single-pass design (cameras, depth, point maps, instance features) is practical, and the instance masks plug neatly into VLM/LMM grounding. The reconstruction results could be comparable with VGGT and in some dataset is better, the semantic results are also good.

4. The idea is simple but efficient and perfroms good.

5. Overall the paper is well-written and the figures are nice.

**Weaknesses:**

1. Some typos:
(1) line 198: " 3) a 3D consistent supervision to" ? To what? Super curious about the following sentence.
(2) In Fig2, I think you are refering to InsScene-15K. But the pie chart and the right column is ScenePart-15K
(3) Scannet --> ScanNet ; Scannet++ --> ScanNet++
2. There are some new baselines in such kind of unified model for semantic understanding. SceneSplat [1,2] trains a large model to take Gaussian Splatting in and open-vocabulary semantic out, and got a great performance on ScanNet and ScanNet++.

[1] SceneSplat: Gaussian Splatting-based Scene Understanding with Vision-Language Pretraining
[2] SceneSplat++: A Large Dataset and Comprehensive Benchmark for Language Gaussian Splatting

3. No outdoor/egocentric/fisheye/low-light results. Can you add some results for the out of distribution data?

**Questions:**

1. Is your model finetuned from VGGT or trained from scratch?
2. A dataset card to detail the strength and uniqueness of InsScene-15K will be good.
3. Please fix the typos in the paper.
4. I saw that all code and checkpoints will be publicly available, will the data also be public?

---

> ### Author Response · Authors · 2025-11-21
>
> ### **W1 & Q3-Typos**
>
> We sincerely thank you for your careful reading and for pointing out these typos. We have corrected all of them in the revised manuscript.
>
> ### **W2-New Baselines**
>
> Thanks for the suggestion. Both SceneSplat and SceneSplat++ are indeed relevant to our work, and we will include a discussion of them in the revised version. These works also focus on 3D scene understanding. However, we would like to clarify the distinct focus of our paper. SceneSplat and SceneSplat++ primarily train a model that takes pre-computed Gaussian Splatting (GS) representations as input to produce open-vocabulary semantic outputs. In contrast, our method focuses on a complete end-to-end (E2E) pipeline, directly mapping multi-view 2D images to a comprehensive 3D understanding and reconstruction. Our model jointly performs 3D reconstruction and understanding, rather than operating on a pre-built 3D representation.
>
> ### **W3-Out-of-Distribution (OOD) Results**
>
> Thanks for the comments. To demonstrate the generalization capability of our model, we evaluated our model on: Outdoor scenes (ETH3D (Schops et al., 2017)), Autonomous driving scenes (Waymo Open Dataset (Sun et al., 2020)), Egocentric-view data (robotics data and a self-collected data). These visualization results are shown in the appendix. As shown, our method is able to correctly handle outdoor scenes (Fig. 19, Fig. 20), low-light scenes (Fig. 21), and even capture short-term dynamic motion in both robotic manipulation and egocentric scenarios (Fig. 22(a)).  In the robotic scenario, the motion of the robotic arm during the grasping sequence is correctly tracked (blue mask), while in the egocentric scenario, the cardboard box is also consistently tracked (blue mask). These diverse test cases demonstrate the robustness of our method, especially considering that it is trained only on static scenes. However, as shown in Fig. 22(b), for the fisheye scenario in egocentric data, our method, similar to VGGT, encounters challenges in producing geometrically accurate reconstructions due to the more complex intrinsic parameters of fisheye cameras.
>
> ### **Q1-Training Detail**
>
> Our IGGT model is finetuned on VGGT model, since it is hard to reproduce the performance on VGGT from scratch.
>
> ### **Q2-Dataset Card**
>
> We provide a dataset card to highlight the uniqueness of our dataset. In Table 1, we evaluate these datasets along five dimensions: RGB images, camera poses, depth, instance masks, and diversity. Datasets with good-quality annotations in a given dimension are marked with "✔", while those without such annotations, or with low-quality ones, are marked with "✖".
>
> Table 1: Comparison of Different Datasets.
>
> | Dataset               | RGB  | Pose | Depth | Instance Masks | Diversity |
> |-----------------------|------|------|-------|----------------|-----------|
> | RE10K                 | ✔    | ✔    | ✖     | ✖              | ✔         |
> | ScanNet               | ✔    | ✔    | ✔     | ✖              | ✖         |
> | ScanNet++             | ✔    | ✔    | ✔     | ✖              | ✖         |
> | Infinigen             | ✔    | ✔    | ✔     | ✔              | ✖         |
> | InsScene-15K (Ours)   | ✔    | ✔    | ✔     | ✔              | ✔         |
>
> ### **Q4-Open Source**
>
> Due to the double-blind review policy, we are unable to provide a URL at this time; however, we commit to releasing all code, checkpoints, benchmarks, and data on Hugging Face after the review process. Meanwhile, we have provided the code in the supplementary material.

---

> > ### Comment · Reviewer_Jjwe · 2025-11-26
> >
> > Thank you for the detailed rebuttal and clarifications. The additional information resolves my earlier concerns. Overall, I find the model design and the proposed dataset sound and well-motivated.

---

> ### Comment · Area_Chair_61CN · 2025-11-25
>
> Dear Reviewer Jjwe,
>
> The authors have responded to your reviews. Please review and provide your feedback and responses.
>
> Best,
>
> Your AC

---

> ### Public Comment · ~Zhirui_Gao2 · 2025-11-27
> **The detail of fineturning the VGGT**
>
> Could you provide more details about fine-tuning the VGGT model? Is it using VGDT parameters directly as initialization, or are some fine-tuning techniques, such as LORA, used?

---

> > ### Author Response · Authors · 2025-11-28
> >
> > The model is directly finetuned from the original VGGT weights without LORA.

---

### Official Review · Reviewer_iRBB · 2025-10-26

**Soundness:** 3
**Presentation:** 3
**Contribution:** 3
**Rating:** 6
**Confidence:** 4

**Summary:**

This paper proposes the Instance-Grounded Geometry Transformer (IGGT) that unifies spatial reconstruction and instance-level contextual understanding in a unified transformer. Using only 2D inputs, IGGT can encode a unified representation with geometric structures and instance-grounded clustering. The authors also construct a dataset named InsScene-15K with instance-level mask annotations.

**Strengths:**

1. I think paper is about the incremental improvements over VGGT by unifying spatial reconstruction and instance-level contextual understanding, which is interesting.
2. The writing and presentation are good, and both quantitative and qualitative experiments have been conducted in sufficient detail.

**Weaknesses:**

1. The discussion of related work is somewhat insufficient. Several recent and more advanced 3D Multimodal LLMs—such as **Inst3D-LMM (CVPR 2025)** and **Chat-Scene (NeurIPS 2024)**—are not discussed. A more comprehensive review would strengthen the paper.
2. Do the authors consider evaluating IGGT on more general 3D scene understanding tasks to further demonstrate its effectiveness—for example, 3D visual grounding on ScanRefer or Multi3DRefer, and 3D VQA on ScanQA?

**Questions:**

Please open-source the code and dataset as soon as possible to enable community validation.
I will also update my assessment after considering feedback and technical insights from other reviewers.

---

> ### Author Response · Authors · 2025-11-21
>
> ### **W1-Related Work**
>
> Thanks for the comments. We have added the discussion of these two papers in the related works. While these works also focus on 3D scene understanding, our method aims at jointly reconstructing and understanding 3D scenes from images, in contrast to their approach of performing 3D grounding via finetuned LMMs operating on point cloud inputs.
>
> ### **W2-Benchmark of QA**
>
> Thank you for the valuable feedback. We agree that evaluating on ScanRefer and ScanQA can help demonstrate the effectiveness of our method for general 3D scene understanding.
> However, it is also worth noting that the original ScanRefer and ScanQA settings assume access to complete room-level point clouds, whereas our method operates only on sparse multi-view images, which is more challenging and more representative of real-world deployment scenarios. Meanwhile, most methods evaluated on these benchmarks rely on specially trained multimodal models (e.g., Chat-Scene and Inst3D-LMM) that are tailored for 3D grounding, whereas our approach is not specifically designed for grounding but instead focuses on unified geometric reconstruction and instance understanding.
> Given these differences in evaluation protocols, we report only qualitative results in this work. As shown in Fig. 18, we showcase two tasks, object counting and spatial relation reasoning, derived from ScanRefer and ScanQA. When provided with the masks generated by our method, Gemini 2.5 Pro successfully completes both tasks with a high success rate. In contrast, the success rate drops significantly without our masks, highlighting the importance and effectiveness of explicit spatial parsing for general 3D scene understanding. In future work, we plan to build a standardized benchmark and conduct comprehensive quantitative evaluations on general 3D scene understanding tasks.
>
> ### **Q1-Open Source**
>
> Due to the double-blind review policy, we are unable to provide a URL at this time; however, we commit to releasing all code, checkpoints, benchmarks, and data on Hugging Face after the review process. Meanwhile, the code has been included in the supplementary material.

---

> > ### Comment · Reviewer_iRBB · 2025-11-21
> >
> > Thanks for the rebuttal, which resolves my primary concerns.
> >
> >  One additional note: the added changes seem to be placed mainly in the Appendix. Since the main paper can be extended to 10 pages during the discussion and camera-ready stages, remember to update the main text accordingly—for example, by adding the newly mentioned papers to *Related Work* and incorporating the new discussions and comparisons into *Experiments*.

---

### Official Review · Reviewer_GKSC · 2025-11-01

**Soundness:** 3
**Presentation:** 3
**Contribution:** 3
**Rating:** 6
**Confidence:** 4

**Summary:**

This work proposes a unified framework that jointly performs 3D reconstruction and class-agnostic instance segmentation from multi-view 2D images. The approach is enabled by introducing a new large-scale dataset that includes RGB images, depth maps, camera poses, and 3D-consistent instance masks, facilitating consistent supervision across views.

To achieve this, the authors design a unified transformer architecture that encodes multi-view RGB images into a shared latent space, from which two task-specific heads decode:
	1.	a Geometry Head, predicting 3D point or depth maps, and
	2.	an Instance Head, producing class-agnostic instance segmentation fields.

To guid the training 3D-Consistent Contrastive Learning strategy, which enforces multi-view consistency in both geometry and instance features by contrasting instance representations across different viewpoints of the same scene.

It also introduces InsScene-15K, a new large-scale dataset that combines both synthetic and real-world data, including RGB images, depth maps, camera poses, and 3D-consistent instance segmentation masks.

Experiments on ScanNet validate the method’s improved performance in 3D reconstruction, instance tracking, and open-world segmentation tasks.

**Strengths:**

The paper addresses an important and timely problem—jointly performing 3D reconstruction and instance-level scene understanding. This capability is highly relevant for downstream applications in robotics, AR/VR, and general 3D scene analysis, where both geometry and object-level understanding are required.

- Producing instance masks alongside reconstruction is meaningful for many robotic and perception tasks, as most downstream reasoning and manipulation systems operate at the instance level rather than on raw pixels or voxels. The proposed framework aligns well with this need.
- The authors contribute a well-curated, large-scale dataset (InsScene-15K) that includes RGB images, depth maps, poses, and 3D-consistent instance annotations. This dataset fills an important gap and could serve as a valuable benchmark for future research.
- The proposed joint learning strategy—where instance-level understanding and geometric reconstruction reinforce each other—is conceptually sound and empirically validated.
- The results showing improvements across multiple downstream tasks (spatial tracking, open-vocabulary segmentation, and scene grounding), demonstrating the model’s effectiveness and generality.

**Weaknesses:**

- An ablation study on the 3D-Consistent Contrastive Loss is missing. Since this loss is central to the paper’s claim of mutual learning between geometry and instance representation, its explicit contribution should be quantified through targeted ablations.
- The paper lacks a standard class-agonistic instance segmentation evaluation. Common metrics such as AP, AP50, and AP25 on ground-truth instance labels—widely used in the 3D instance segmentation literature (e.g., SAI3D, SamPart3D, OpenIns3D)—would provide a more rigorous assessment of instance mask quality. Given that instance quality directly impacts downstream tasks such as scene grounding and spatial tracking, this omission weakens the quantitative validation.
- 3D instances can often be extracted directly from point clouds using clustering or graph-based grouping methods (e.g., VGGT + standard 3D clustering (SuperPoint or Graph Cut). Therefore, evaluating the quality of class-agnostic instance masks against such baselines would strengthen the paper’s claim that joint learning between geometry and instance representation is beneficial.
- A runtime analysis could be provided.

**Questions:**

- The quality of the instance labels obtained from the network is crucial part of the contribution. More information on this aspect would be necessary to confirm the reliability of the results.

- The definition of the Spatial Tracking task is not entirely clear. Spatial trackers typically refer to temporal tracking in dynamic videos, where both scene motion and camera motion coexist. In static multi-view setups, pixel correspondences are largely determined by camera movement. From the visualizations, it appears to refer to instance matching across views? More clarity on this point would be helpful.

- How is the granularity of the instance masks determined? Is this property controllable during inference or training?

---

> ### Author Response · Authors · 2025-11-21
>
> ### **W1-Ablation of 3D-Consistent Contrastive Supervision**
>
> Thank you for the comments. As shown in Table 1, we perform an ablation study on ScanNet++ dataset regarding the weighting coefficient $\lambda$ of the contrastive loss $L_{\text{mvc}}$, with the weight of the geometric losses fixed to $1$. Specifically, we scale $\lambda$ by factors of $\times 2$ and $\times 10$ to analyze its impact on performance. Here, our method maintains comparable performance under the $\times 2$ setting (i.e.,  $\lambda = 0.5, 2$ ), demonstrating the robustness of the overall training procedure. In contrast, under the $\lambda = 0.1$ setting, the contrastive supervision fails to provide sufficiently discriminative instance features, leading to degraded mask quality in the post-processing clustering stage, resulting a big drop in understanding performance (10.44\% in 2D mIoU). Meanwhile, under the $\lambda = 10$ setting, the contrastive supervision becomes dominant and weakens the geometric supervision, leading to degraded geometric reconstruction performance (a 6.28\% drop in $\tau$) as well as a decline in 3D segmentation accuracy (a 4.96\% drop in 3D mIoU).
>
> Table 1: Ablation study with different values of $\lambda$ of contrastive supervision
> | Metrics | $\lambda = 0.1$ | $\lambda = 0.5$ | $\lambda = 1$ | $\lambda = 2$ | $\lambda = 10$ |
> |---------|------------------|------------------|----------------|----------------|-----------------|
> | 2D mIoU | 20.63 | 31.07 | 31.31 | 31.86 | **32.60** |
> | 3D mIoU | 13.89 | 19.79 | **20.14** | 20.01 | 15.18 |
> | $\tau$  | 85.35 | 85.43 | **85.66** | 85.19 | 79.38 |
>
> ### **W2 & W3 & Q1-Class-Agnostic Evaluation**
>
> Thanks for the valuable comments. We have conducted class-agnostic instance segmentation experiments on the ScanNet dataset, comparing our method against VGGT + Graph Cut and VGGT + SAI3D.
>
> **Settings:** For VGGT + Graph Cut, we first use VGGT to reconstruct the entire scene as a point cloud from the input images, then compute normal vectors for the points and generate a mesh using Poisson reconstruction. Finally, following SAI3D, we apply a normal-based graph cut algorithm to over-segment the mesh into segmentation results (superpoints). For VGGT + SAI3D, the pipeline is directly built on VGGT + Graph Cut, and the predicted depth, poses, and superpoints are then fed into SAI3D to obtain the final class-agnostic instance segmentation results.
>
> **Evaluation Protocol:** We follow the ScanNet instance segmentation benchmark to evaluate class-agnostic instance segmentation. First, we use the ground-truth depth and camera poses to project the ground-truth 2D instance masks into point clouds, obtaining the ground-truth class-agnostic 3D instance masks. Subsequently, we align the point clouds predicted by IGGT and VGGT with the ground-truth point clouds, and then apply nearest-neighbor matching to establish a one-to-one correspondence between the predicted and ground-truth class-agnostic 3D instance masks. Additionally, we followed the practice of SAI3D by ignoring instances representing 'wall' and 'floor'.
>
> **Results:** We report the average precision scores at IoU thresholds of 0.25 (AP25) and 0.50 (AP50), as well as averaged over IoU thresholds from 0.50 to 0.95 in increments of 0.05 (AP) in Table 2. Our method significantly outperforms graph-based grouping approaches such as VGGT+Graph Cut across all metrics, achieving an 8.83 improvement in AP. We provide a qualitative comparison in Fig. 16. Meanwhile, since this graph cut algorithm requires a mesh as input, a substantial amount of time is spent on mesh generation, whereas our approach can directly output multi-view consistent instance features for clustering, reducing the overall runtime by approximately 8 minutes. Moreover, our method even approaches the performance of the per-scene optimization method VGGT+SAI3D, while achieving nearly a 5$\times$ reduction in runtime (2.5 min vs. 12.2 min). This further showcases the effectiveness of our unified framework, which jointly learns geometric reconstruction and instance understanding.
>
> Table 2: Comparison of different methods on AP metrics
>
> | Method            | AP    | AP$_{50}$ | AP$_{25}$ | Time    |
> |-------------------|--------|-----------|-----------|---------|
> | VGGT + Graph Cut  | 3.42   | 9.30      | 30.86     | 10.65min |
> | VGGT + SAI3D      | 14.94  | 31.06     | 50.07     | 12.22min |
> | Ours              | 12.25  | 24.93     | 47.55     | 2.52min  |

---

> ### Author Response · Authors · 2025-11-21
>
> ### **W4-Runtime Analysis**
>
> We appreciate your suggestion. Here, we use 10 images from a single scene to evaluate the detailed runtime, as reported in Table 3.
>
> **Open Vocabulary Semantic Segmentation**: For NeRF and 3DGS methods, they require massive time consumption for per-scene optimization. Moreover, they also require camera poses as input, which are typically estimated using COLMAP. For the LSM (Multi-Views) method, building on its two-view inference, we further feed all images into Dust3R to infer the camera parameters and point maps for each view, and then apply the global alignment strategy of Dust3R to optimize the point maps across all views. Although it takes only about 30 seconds, its open-vocabulary semantic segmentation performance is inferior to ours, and it is unable to produce class-agnostic instance segmentation results.
>
> **Class-Agnostic Instance Segmentation**: For the VGGT+Graph Cut method, we adopt the normal-based graph cut algorithm provided by the ScanNet official code. We compute normal vectors for the point clouds predicted by VGGT and generate a mesh using Poisson reconstruction. These additional steps are time-consuming and introduce significant overhead. For the VGGT+SAI3D method, which is built upon the VGGT+Graph Cut pipeline, a substantial portion of the runtime is similarly spent on mesh processing. For our method, most of the runtime is spent on the HDBSCAN clustering step, which takes about two minutes. Nevertheless, our approach remains the second fastest among all compared methods.
>
> Table 3: Comparison of runtimes for different methods
>
> | Method            | SfM Time | Per-Scene Time | Infer. Time | Post Proc. Time | Final Time |
> |-------------------|----------|----------------|------------|----------------|------------|
> | NeRF-DFF           | 50.33s  | 3min           | -          | -              | 3.84min   |
> | Feature-3DGS       | 50.33s  | 47min          | -          | -              | 47.84min  |
> | LSM (Multi-Views)  | -       | -              | 15.98s     | 13.72s         | 29.70s    |
> | VGGT+Graph Cut     | -       | -              | 0.426s     | 10.64min       | 10.65min  |
> | VGGT+SAI3D         | -       | -              | 0.426s     | 12.21min       | 12.22min  |
> | Ours               | -       | -              | 0.545s     | 2.51min        | 2.52min   |
>
> ### **Q2-Definition of Spatial Tracking Task**
>
> "Multi-View Instance Matching" is a more accurate description of our task, since our current benchmark only involves camera motion in a static scene. Accordingly, we have replaced "Spatial Tracking" with "Multi-View Instance Matching" in the revision.
>
> ### **Q3-Granularity**
>
> Since our method does not support prompt-based inputs like SAM or Semantic-SAM, it is unable to directly predict instance features at different granularities. However, we can still obtain instance masks at different scales by adjusting the parameters of HDBSCAN. As illustrated in Fig. 17, we showcase three different clustering granularities, where the chair serves as the discriminative region of interest. Nevertheless, this remains an important direction for future work, potentially involving improved model designs (e.g., DETR-based architectures) and more scalable datasets, such as SA-1B dataset used in SAM1 and Semantic-SAM, and SA-V dataset used in SAM2.

---

> ### Comment · Area_Chair_61CN · 2025-11-25
>
> Dear Reviewer GKSC,
>
> The authors have responded to your reviews. Please review and provide your feedback and responses.
>
> Best,
>
> Your AC

---

### Author Response · Authors · 2025-12-02
**Summary Message.**

We sincerely thank all reviewers, AC, SAC, and PC for their precious time, thoughtful feedback, and engagement throughout the review period.

Firstly, we would like to once again highlight the key motivation and novelty of this paper. We hope this helps in better understanding of our work:

**Motivation and Novelty:**

1. Unlike prior approaches that treat reconstruction and understanding tasks in isolation or simply align with specific language models, we propose **Instance-Grounded Geometry Transformer (IGGT)**, an end-to-end large unified transformer to unify the knowledge for both spatial reconstruction and instance-level contextual understanding.
2. We design a 3D-Consistent Contrastive Learning strategy that guides IGGT to encode a unified representation with geometric structures and instance-grounded clustering through only 2D visual inputs.
3. We further construct InsScene-15K, a large-scale dataset with high-quality RGB images, poses, depth maps, and 3D consistent instance-level mask annotations with a novel data curation pipeline.
4. We introduce Instance-Grounded Scene Understanding to support various applications, including multi-view instance matching, class-agnostic mask segmentation, open-vocabulary segmentation, and QA scene grounding.

Secondly, we have submitted a revised manuscript that fully incorporates the reviewers’ feedback and includes additional experiments conducted during the rebuttal. All revised sections in the manuscript are highlighted in $\textbf{\textcolor{blue}{blue}}$ for easy reference. In summary:

**1. Supplementary Ablation Studies:**
* Conducted ablation of the 3D-Consistent Contrastive Loss weighting coefficient ($\lambda$) on the ScanNet++ dataset in **Sec. A.6 and Tab. 7** (Reviewer GKSC).
* Conducted ablations across different segmentation granularities in **Sec. A.9 and Fig. 19** (Reviewer GKSC).

**2. New Class-Agnostic Instance Segmentation Evaluation:**
* Compared our method against **VGGT+Graph Cut** and **VGGT+SAI3D** on the ScanNet dataset using AP metrics ($AP_{25}$, $AP_{50}$, $AP$) in **Sec. 4, Sec. A.7, Table 3, Fig. 9, Fig. 18** (Reviewer GKSC).

**3. Detailed Runtime Analysis:**
* Provided a comprehensive runtime comparison table across all methods in **Sec. 4, Sec. A.8 and Fig. 12** (Reviewer GKSC).

**4. Detailed VQA Analysis:**
* Provided a comprehensive discussion between our method and existing LLM-based methods, and further conducted qualitative demonstrations on ScanRefer and ScanQA in **Sec. A.10, Fig. 20** (Reviewer iRBB).

**5. Out-of-Distribution (OOD) Evaluation:**
* Conducted additional visualization on OOD scenarios, including outdoor, autonomous driving, egoview, robotics, and low-light scenes in **Sec. 4, Sec. A.11 and Fig. 10, Fig. 21, Fig. 22, Fig. 23, Fig. 24** (Reviewer Jjwe).

**Summary of the reviewer discussion period:**

Firstly, all reviewers provided positive feedback and ratings. After our item-to-item response, both **reviewers iRBB and Jjwe** indicated that their primary concerns had been fully addressed.
Regarding **GKSC**, although no additional comments were provided before the OpenReview leaking accident, we conducted thorough experiments on class-agnostic segmentation and showcased IGGT’s strong performance, which we believe adequately addresses their primary concern.

---

### Meta-Review · Area_Chair_tFPY · 2025-12-22

**Summary:**

Reviewers generally find the idea solid and the dataset potentially useful, but the paper initially had some gaps in evaluation and positioning. The main concerns were missing/weak evidence around the role of the contrastive objective, lack of standard class-agnostic instance segmentation metrics and relevant baselines, missing runtime evaluation, and insufficient coverage of closely related 3D grounding/VLM work.

**Reviewer Concerns:**

The authors added an ablation for the contrastive loss, reported class-agnostic instance segmentation AP-style metrics with baselines (VGGT+GraphCut, VGGT+SAI3D), provided a runtime table, and clarified that “spatial tracking” is really multi-view instance matching. They also expanded related work, added qualitative grounding/VQA examples, and included additional OOD visualizations, plus fixed typos and clarified that they fine-tune from VGGT.

**Reviewer Scores:**

All reviewers are likely to maintain or increase their scores, provided that their concerns are properly addressed or explained.

---

### Decision · Program_Chairs · 2026-01-26

Accept (Poster)